# Phase of firing coding of learning variables across the fronto-striatal network during feature-based learning

Benjamin Voloh[1], Mariann Oemisch[1] & Thilo Womelsdorf [1✉]

The prefrontal cortex and striatum form a recurrent network whose spiking activity encodes multiple types of learning-relevant information. This spike-encoded information is evident in average firing rates, but finer temporal coding might allow multiplexing and enhanced readout across the connected network. We tested this hypothesis in the fronto-striatal network of nonhuman primates during reversal learning of feature values. We found that populations of neurons encoding choice outcomes, outcome prediction errors, and outcome history in their firing rates also carry significant information in their phase-of-firing at a 10–25 Hz band-limited beta frequency at which they synchronize across lateral prefrontal cortex, anterior cingulate cortex and anterior striatum when outcomes were processed. The phase-of-firing code exceeds information that can be obtained from firing rates alone and is evident for inter-areal connections between anterior cingulate cortex, lateral prefrontal cortex and anterior striatum. For the majority of connections, the phase-of-firing information gain is maximal at phases of the beta cycle that were offset from the preferred spiking phase of neurons. Taken together, these findings document enhanced information of three important learning variables at specific phases of firing in the beta cycle at an inter-areally shared beta oscillation frequency during goal-directed behavior.

[1] Department of Psychology, Vanderbilt University, Nashville, TN, USA. ✉email: thilo.womelsdorf@vanderbilt.edu

The lateral prefrontal cortex (LPFC) and anterior cingulate cortex (ACC) are key brain regions for adjusting to changing environmental task demands[1,2]. Both regions project to partly overlapping regions in the anterior striatum (STR), which feeds back projections via the thalamus and thereby close recurrent fronto-striatal-thalamic loops[3]. Neurons in this recurrent network encode multiple learning variables during goal-directed behaviors, including the value of currently received outcomes, a memory of recently experienced outcomes, and a reward prediction error that indicates how unexpected currently received outcomes were given prior experiences[4,5].

The multiplexing of outcomes, outcome history and outcome unexpectedness (prediction errors) within the same neuronal population is evident in firing rate modulations in fronto-striatal brain areas[6], but how this firing is temporally organized within the larger network is unresolved[7–9]. A large body of evidence has shown that ACC and LPFC synchronize their local activities at a characteristic beta oscillation frequency[10–13], and that both areas engage in transient beta rhythmic oscillatory activity with the STR during complex goal-directed tasks[14–17]. However, whether this beta oscillatory activity is informative for learning and behavioral adjustment has remained unresolved[18–20]. Prior studies have documented that beta activity emerges specifically during the processing of outcomes following correct trials during habit learning[17], and that, following error trials, overall beta activity is larger when the committed error is smaller[21]. However, these studies did not quantify whether neuronal spiking activity synchronizing to these beta oscillations contains learning-relevant outcome information.

We, therefore, aimed to test how outcome-related beta rhythmic spiking activity relates to the behavioral learning of reward rules in ACC, LPFC, and STR. First, we quantified firing rate information about current outcomes, prediction errors of these outcomes, and the history of recent reward. These variables might be conveyed independently of network-level beta oscillatory activity. However, theoretical studies suggest that neuronal

coding utilizing temporal organization can be efficient, high in capacity, and robust to noise[7,8,22,23]. In addition, coding of information in the temporal activity pattern has been linked to mechanisms of efficient communication among neuronal groups, suggesting that coherently synchronized groups can exchange information by phase aligning their disinhibited activity periods[24–30].

To test the role of temporal coding, we recorded from LPFC, ACC, and STR while macaque monkeys engaged in trial-and-error reversal learning of feature reward rules. We found that during outcome processing, each area contains segregated ensembles of neurons whose firing rates encode the current Outcome (firing differently for correct vs. errors), the Reward Prediction Error of those outcomes (firing differently to an outcome when it differed versus was the same than in the previous trial, as in e.g.,[31,32]), and the recent Outcome History (increasing firing when the current outcomes matched previous outcomes). A large proportion of rate coding neurons phase-synchronized long-range to remote areas of the fronto-striatal network at a shared 10–25 Hz beta frequency range. We found that for those neurons that phase-synchronize long-range, the three learning variables are encoded more precisely for spikes elicited at narrow oscillation phases in the beta band. This phase-of-firing gain of encoding significantly enhances the firing rate code and occurs at phases that were partly away from the neurons' preferred spike phase. These findings document that neural coding of learning variables is enhanced through the phase of firing across the ACC, LPFC, and STR of nonhuman primates.

## Results

**Previous outcomes guide choice.** Animals performed a feature-based reversal-learning task[5]. Subjects were shown two stimuli with opposite colors and had to learn which of them led to reward (Fig. 1a). The same color remained associated with reward for at least 30 trials before an uncued reversal switched the color-reward association (Fig. 1b). During each trial, the subjects monitored the

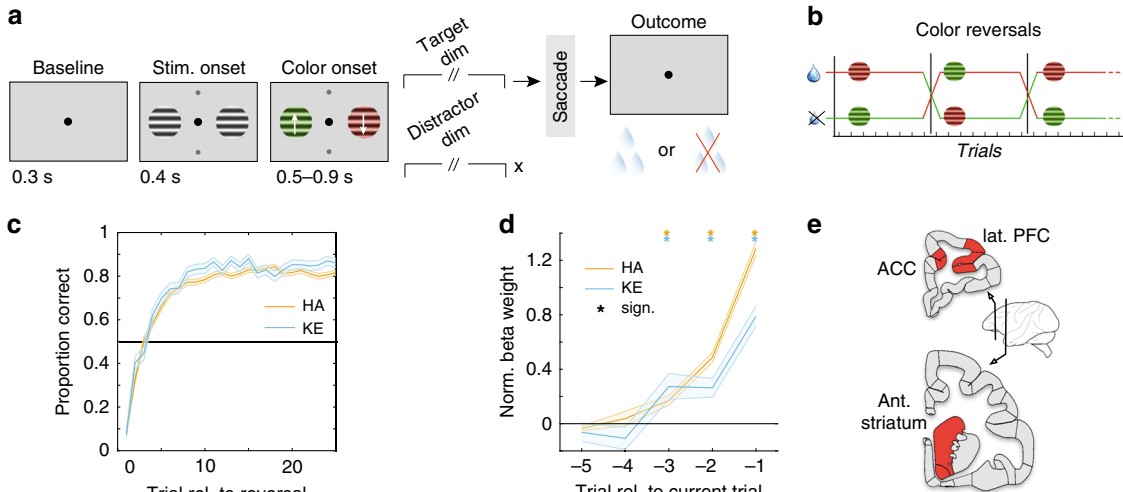

**Fig. 1 Task design and performance. a** Feature-based reversal-learning task. Animals are presented with two black/white stimulus gratings to the left and right of a central fixation point. The stimulus gratings then become colored and start moving in opposite directions. Dimming of the stimuli served as a Go signal. At the time of the dimming of the target stimulus the animals had to indicate the motion direction of the target stimulus by making a corresponding up or downward saccade in order to receive a liquid reward. Dimming of the target stimulus occurred either before, after or at the same time as the dimming of the distractor stimulus. **b** The task is a deterministic reversal-learning task, whereby only one color is rewarded in a block. This reward contingency switches repeatedly and unannounced in a block-design fashion. **c** Accuracy relative to block start for monkey HA (orange) and KE (blue). The shaded region represents the standard error. Subjects achieved plateau performance within 5–10 trials **d** Median beta coefficients from a binomial regression of current outcome as predicted by past outcomes. The shaded region represents the standard error. Outcomes up to three trials into the past predicted current outcome (Wilcoxon signrank, two-sided multiple comparison corrected; stars represent $p < 0.05$). **e** Schematic depicting recorded brain areas.

stimuli for transient dimming events in the colored stimuli. They received a fluid reward when making a saccade in response to the dimming of the stimulus with the reward associated color, while the dimming of the non-reward associated color had to be ignored. A correct, rewarded saccade to the dimming of the rewarded stimulus had to be made in the up- or downward direction of motion of that stimulus. This task required covert selective attention to one of two peripheral stimuli based on color, while the overt choice was based on the motion direction of the covertly attended stimulus. Correct responses were those that occurred according to the motion direction of the rewarded target in the correct response window, whereas errors were responses made to the incorrect, non-rewarded target, or in the incorrect response window in response to the distractor[5]. In 110 and 51 sessions from monkey's HA and KE, respectively, we found that subjects attained plateau performance of on average 80.2% (HA: 78.8%, KE: 83.6%) within 10 trials after color-reward reversal (Fig. 1c). Using a binomial General Linear Model (GLM) to predict current choice outcomes (outcomes from correct or erroneous choices, excluding fixation breaks), we found that for both subjects, outcomes from up to three trials into the past significantly predicted the current choice's outcome (Fig. 1d; Wilcoxon signrank test, $p < 0.05$, multiple comparison corrected), closely matching previous findings[33].

**ACC, LPFC, and STR neurons encode outcomes, their history and their prediction error**. To test how previous and current outcomes are encoded at the single neuron level, we analyzed a total of 1460 neurons, with 332/227 (monkey HA/KE) neurons in LPFC, 268/182 neurons in ACC, and 221/230 neurons in anterior STR (Fig. 1e, Supplementary Fig. 1). These regions have previously been shown to encode outcome, outcome history, and prediction error information[5,34–39]. We found multiple example neurons encoding different types of outcome variables. Some cells responded differently to correct versus erroneous trial outcomes irrespective of previous outcomes (Fig. 2a), while others responded strongest when the current outcome deviated from the previous trials' outcome (signaling reward prediction error) (Fig. 2b), or when the current outcome was similar to the previous trials' outcome, i.e., following a sequence of correct trials or a sequence of error trials (Fig. 2c).

We quantified these types of outcome encoding using a LASSO Poisson GLM that predicted the spike counts during the outcome period (0.1–0.7 s after reward onset) and extracted the characteristic patterns of beta weights across the past and current trial outcomes that distinguished different types of outcome encoding (Fig. 2d). Neurons that encoded mostly the current trials' outcome showed large weights only for the current trial (Outcome encoding type). Neurons encoding a prediction error showed beta weights for previous trials that were opposite in sign to the current trial's outcome (Reward Prediction Error (RPE) encoding type). In neurons encoding the history of recent rewards, beta weights ramped up over recent trials toward the current trial outcome (Outcome History encoding type) (for examples, see also insets in Fig. 2a–c).

We used a clustering analysis to test whether the three types of outcome encoding were separable from each other and prevalent in each of the recorded brain areas (Fig. 2e–g, Supplementary Fig. 2A, B). Clustering showed that neurons encoding each the three variables were statistically separable with reliable cluster assignments of neurons evident in an average Silhouette measure of cluster separability of 0.81 for LPFC, 0.57 for ACC, and 0.75 for STR (Supplementary Fig. 2C)[40]. The clustering does not preclude the possibility of a more continuous encoding space, but it statistically justifies focusing analysis on three sets of neurons

with well distinguishable encoding pattern (see Supplementary Fig. 2D).

Across the population the proportion of neurons with significant encoding, Outcome cells were the most populous (~59%, 234/384 in monkey HA and 185/329 in monkey KE), followed by ~26% of neurons encoding Reward Prediction Errors (64/231 in monkey HA and 39/170 in monkey KE) and ~32% of neurons encoding Outcome History (76/206 in monkey HA and 33/139 in monkey KE) (Fig. 2e–g; $\chi^2$ test, $\chi^2 = 86.02$, $p\sim0$). The relative frequency of these encoding types did not differ between areas ($\chi^2$ test, $\chi^2 = 3.64$, $p = 0.46$). On the other hand, the strength of encoding differed on the basis of area for Outcome cells (Kruskal–Wallis test, $\chi^2 = 26.6$, $p\sim0$), with stronger encoding in ACC than LPFC or STR, as well as for Outcome History encoding cells ($\chi^2$-test, $\chi^2 = 19.7$, $p\sim0$) with stronger encoding in ACC and LPFC than in STR, whereas the strength of *RPE* encoding was similar across areas ($\chi^2 = 2.49$, $p = 0.29$) (see Supplementary Fig. 2E for all pair-wise comparisons). In ACC, LPFC and STR, Outcome, RPE and Outcome History encoding emerged shortly (within 0.3 s) after outcomes were received (see "Methods"; Supplementary Fig. 2G, H, Wilcoxon signrank test, p«0.05). Neurons encoding Outcome, RPE, or Outcome History showed similar overall firing rates (Supplementary Fig. 2F; ANOVA; LPFC, $F = 1.32$, $p = 0.27$; ACC, $F = 0.58$, $p = 0.58$; STR, $F = 1.05$, $p = 0.35$).

**Neurons synchronize at a 10–25 Hz beta band across ACC, LPFC, and STR**. We found similar proportions and activation time courses of encoding neurons in ACC, LPFC and STR (Supplementary Fig. 2G, H), which raised the question how these neuronal populations are functionally connected. One possibility is that neuronal firing patterns are organized temporally, such that spikes in one area phase synchronized to neuronal population activity in remote areas. We assessed synchrony as the phase consistency of neuronal spikes with local field potential (LFP) fluctuations in distally recorded areas using the pairwise-phase consistency (PPC) metric, and converting the PPC values into an effect size[41,42] (Fig. 3a; see "Methods"). Across all ($n = 7938$) spike-LFP pairs, we found a pronounced peak of phase synchronization in the beta band (10–25 Hz), with neurons firing on average ~1.15 times more spikes on their preferred, average phase than at the opposite phase when considering the population average in the beta band (Fig. 3b), and ~1.39 times more spikes on the preferred phase when selecting for each neuron the beta frequency with peak synchrony (Supplementary Fig. 3A). Prominent beta-band synchrony was evident for neurons that encoded outcome variables in their firing rates and for those that did not show encoding (Fig. 3b), with the peak synchrony being stronger for cell-LFP pairs with non-coding rather than coding cells (unpaired $t$-test, $T = 7.67$, $p\sim0$; Supplementary Fig. 3A). Overall 53% (4230/7938) of the spike-LFP pairs showed significant phase synchronization within the 10–25 Hz range (Fig. 3c; Rayleigh test, $p < 0.05$, see "Methods" for prominence criteria), with similar proportions across all three areas (LPFC, 1506/2961, 50.9%; ACC, 1473/2442, 60.3%; STR, 1292/2524, 51.2%).

Consistent with these results we found that the synchrony effect (the proportion of spikes at preferred over non-preferred phases) were similarly high for spike-LFP pairs with neurons encoding Outcome (1.37 ± 0.007), RPE (1.35 ± 0.013), and Outcome History (1.34 ± 0.011). There was only a trend for phase synchronization to be different between encoding clusters (ANOVA, $F = 2.8$, $p = 0.061$), which post-hoc analysis revealed to be driven primarily by differences between Outcome History and Outcome clusters ($p = 0.078$, multiple comparison

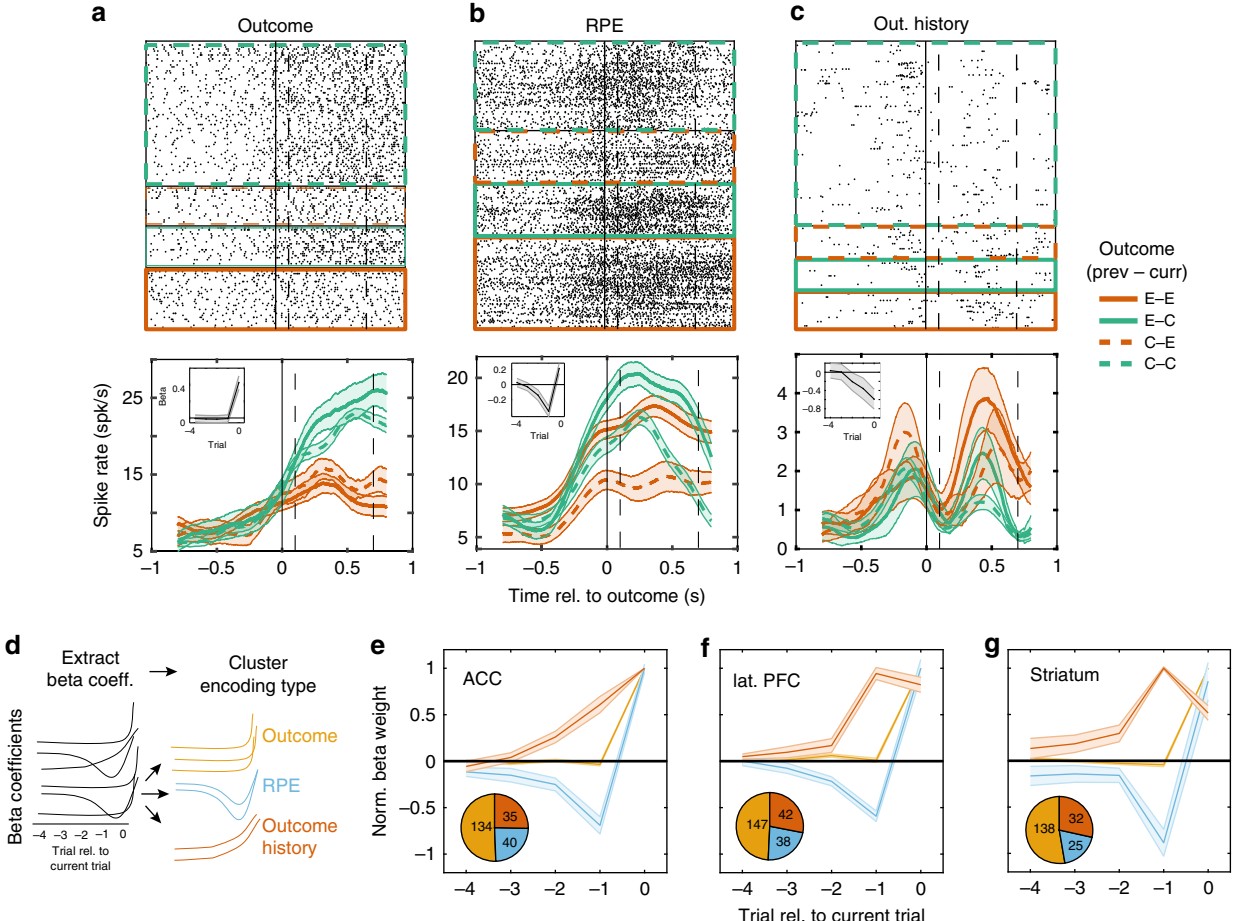

**Fig. 2 Outcomes, prediction errors, and outcome history is encoded across the fronto-striatal axis. a–c** Examples of Outcome (**a**), Reward Prediction Error (RPE)*;* (**b**), and Outcome History (**c**) cells. (top) Raster plot for four separate trial conditions. Color denotes outcome on the current trial (green = correct, red = incorrect), and line style to the previous trial (solid = error, dashed = correct). Previous/current trial pairs are depicted in the legend, where "C" is correct trials, "E" is error trials. The dotted vertical lines represent the time period used for the GLM analysis. (bottom) Time-resolved firing rate for the four different trial types. Insets in the top left depict the average GLM beta coefficients. **d** Sketch of general approach. We regressed spike counts $C$ in the [0.1 0.7] outcome period against outcomes up to 5 trials into the past using a penalized LASSO GLM. Beta coefficients were then clustered to group cells according to their most parsimonious functional designation. **e–g** Median normalized beta coefficients for three functional clusters in ACC (**e**), LPFC (**f**), and STR (**g**). Shaded regions represent the standard error. On the basis of the pattern of beta weights (see text), each area exhibits "Outcome" cells (yellow), "Outcome History" cells (red), and "RPE" cells (blue). (inset) Relative frequency of each functional cluster. Outcome cells were the most populous in all three regions (Chi-squared test, $p \ll 0.05$).

corrected), rather than Outcome History and RPE ($p = 0.80$) or RPE and Outcome clusters ($p = 0.37$).

Next, we tested whether spike-LFP synchronization showed area-specificity for neurons that significantly encoded Outcome, RPE or Outcome History in their firing rates. For each spike-LFP pair, we selected the beta band frequency with the most prominent PPC value. Within-area beta synchrony differed on the basis of area (ANOVA, $F = 32.6$, $p{\sim}0$), with the strongest synchrony within ACC, compared to LPFC (multiple comparison corrected, $p = 0.014$) or STR ($p{\sim}0$) (Fig. 3d, e). Synchrony also differed when assessing spikes and LFPs from different areas (ANOVA, $F = 12.7$, $p{\sim}0$), with neurons in ACC showing stronger between-area spike-LFP synchrony, as compared to LPFC ($p{\sim}0$) and STR ($p = 0.042$). We found a trend for stronger between-area synchrony with spikes originating in STR, as compared to LPFC ($p = 0.058$) (Fig. 3d, f). Testing for the reciprocity of beta-band phase synchrony showed that ACC spikes phase synchronized more strongly to LFP beta activity in the LPFC than vice versa ($p = 0.047$) (Fig. 3g). LPFC and STR pairs showed statistically indistinguishable spike-phase synchrony strength ($p = 0.92$), as did ACC and STR pairs ($p = 0.26$). The findings were similar

when inter-areal synchrony was analyzed separately at each frequency (Supplementary Fig. 3b). For both monkeys, neurons in ACC showed the strongest spike synchronization compared to neurons from LPFC and STR (the area difference is significant in monkey HA, and trends the same way in monkey KE; see Supplementary Fig. 4A). Moreover, across all three areas, the strength of 15–25 Hz phase synchronization was statistically indistinguishable in the [−1 0] s. pre-outcome period compared to the [0.1 1] s. post-outcome period (paired $t$-test, abs($T$) < 1.57, $p > 0.12$; Supplementary Fig. 3c). The baseline period ([−1 0] before stimulus onset) and the post-outcome period showed similar PPC values in ACC and STR (abs($T$) <0.49, $p > 0.6$), while LPFC showed stronger phase synchronization in the post-outcome period ($T = 2.82$, $p = 0.0049$; Supplementary Fig. 3C).

**Phase-of-firing at 10–25 Hz enhances encoding outcome, prediction error and outcome history.** Neurons that synchronized to the LFP elicit more spikes at their mean spike-LFP phase, which we denote as the neurons' preferred spike phase[30]. This preferred spike-phase might thus be important to encode

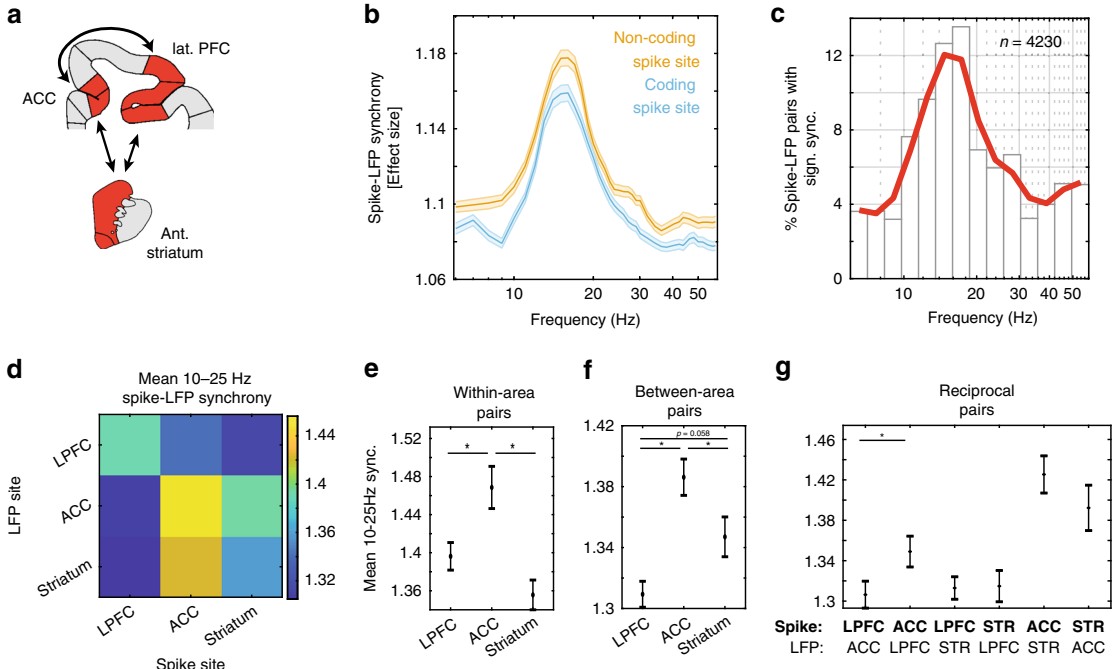

**Fig. 3 Spike-LFP phase synchronization. a** Illustration of the regions of interest in the ACC, LPFC, and STR (in red) depicted on coronal brain sections of the cortex and striatum. **b** Average PPC effect size between distal LFP and spikes at encoding (blue) and non-encoding (orange) spike sites. The shaded area represents the standard error. There is a prominent beta peak for both. The X-axis is depicted on a log-scale for clarity. **c** The proportion of spike-LFP pairs that exhibited significant, prominent locking. Pairs could contribute to more than one bin if they locked to multiple frequencies. (see "Methods"). **d** Mean inter-areal synchrony between all pairs of areas for signification, prominent beta in the [10–25] Hz range. Color denotes the average. **e** Mean and standard error contrasting spike-phase locking within different areas (diagonal in Fig. 3d). ACC synchronizes more strongly that either LPFC or STR. (ANOVA with post-hoc test, multiple comparison corrected; $n_{LPFC} = 348$, $n_{ACC} = 305$, $n_{STR} = 444$). (**f**) Same as (**e**) but for LFPs originating in other areas (i.e., summing across columns, less the diagonals, in Fig. 3d). ACC synchronizes more strongly to distal beta, compared to LPFC or STR. ($n_{LPFC} = 829$, $n_{ACC} = 890$, $n_{STR} = 647$). **g** All pairwise comparison between regions. ACC spikes lock more strongly to LPFC beta than the inverse. ($n_{LPFC-ACC} = 314$, $n_{ACC-LPFC} = 457$, $n_{LPFC-STR} = 515$, $n_{STR-LPFC} = 378$, $n_{ACC-STR} = 433$, $n_{ASTR-ACC} = 269$). Stars represent $p < 0.05$.

information shared among areas of the network[24,43]. We tested this hypothesis by quantifying how much Outcome, RPE, and Outcome History information is available to neurons at different phase bins relative to their mean phase. If the phase-of-firing conveys information, then differences in spike counts between conditions should vary across phases, as opposed to a pure firing rate code that predicts equal information for spike counts across phase bins (Fig. 4a)[44,45]. Figure 4b shows an example neuron exhibiting such phase-of-firing coding (with spikes from ACC and LFP beta phases from STR). This neuron exhibited increased firing on error trials compared to correct trials, but only when considering spikes near its preferred spike phase, with firing at the opposite phase showing no difference. To quantify this increase of coding when considering the phase-of-firing code for all three information types, we selected for each neuron the frequency within 10–25 Hz that showed maximal spike-LFP synchrony, subtracted the mean (preferred) spike phase from all phases (i.e., setting the preferred phase to zero, to allow for comparison between neurons), and binned spikes on the basis of the LFP beta phases. To prevent an influence of overall firing rate changes between phase bins, we adjusted the width of each of the six-phase bins to have equal spike counts across bins (see "Methods"). We then fitted a GLM to the firing rates of each phase bin separately to quantify the Outcome, RPE, and Outcome History encoding for each phase bin and compared this phase specific encoding to a null distribution obtained by randomly shuffling the spike phases prior to binning. Figure 4c illustrates example neurons for which the encoding systematically varied as a function of phase (for more examples, see Supplementary Fig. 5). The example spike-LFP pair from Fig. 4b encoded the trial Outcome

significantly stronger than a phase-blind rate code in spikes within $\sim[-\pi/2, \pi/2]$ radians relative to its preferred spike phase and weaker than a phase-blind rate code at opposite phases (Fig. 4c, left); Enhanced phase-of-firing encoding was similarly evident for RPE and Outcome History as independent variable (Fig. 4c, middle and right panels).

We estimated the strength of this phase modulation of rate encoding for each spike-LFP pair as the amplitude of a cosine that was fit to the phase-binned encoding metric, normalized by the mean encoding across phase bins, which we term the Phase-of-Firing Gain (PFG). We further accounted for the positive bias in cosine amplitude estimation by normalizing this quantity by the cosine amplitude obtained from fitting the phase-binned metric after randomly shuffling spike phases. We refer to this difference of the observed to the randomly shuffled phase modulation of encoding as the Encoding Phase-of-Firing Gain (EPFG; see "Methods"). This metric reflects an unbiased ratio of firing rate differences between preferred and anti-preferred encoding-phase bins. Of the 876 spike-LFP pairs that significantly synchronized in the 10–25 Hz band and encoded information in their firing rate, we found that 139 (16%) spike-LFP pairs showed significant phase-modulation, i.e., these pairs encoded significantly more information when taking into account the phase of firing than their average, phase-blind firing rate (randomization test, $p < 0.05$). A significant EPFG was evident for neurons whose firing encoded Outcome (Wilcoxon signrank test, $Z = 8.41$, p~0), RPE, ($Z = 3.27$, $p = 0.011$), and Outcome History ($Z = 3.24$, $p = 0.012$). The EPFG did not differ between these three functional clusters (Kruskal–Wallis test, $\chi^2 = 0.283$, $p = 0.87$) (Fig. 4d). Similarly, EPFG was evident for spike-LFP pairs when the spiking

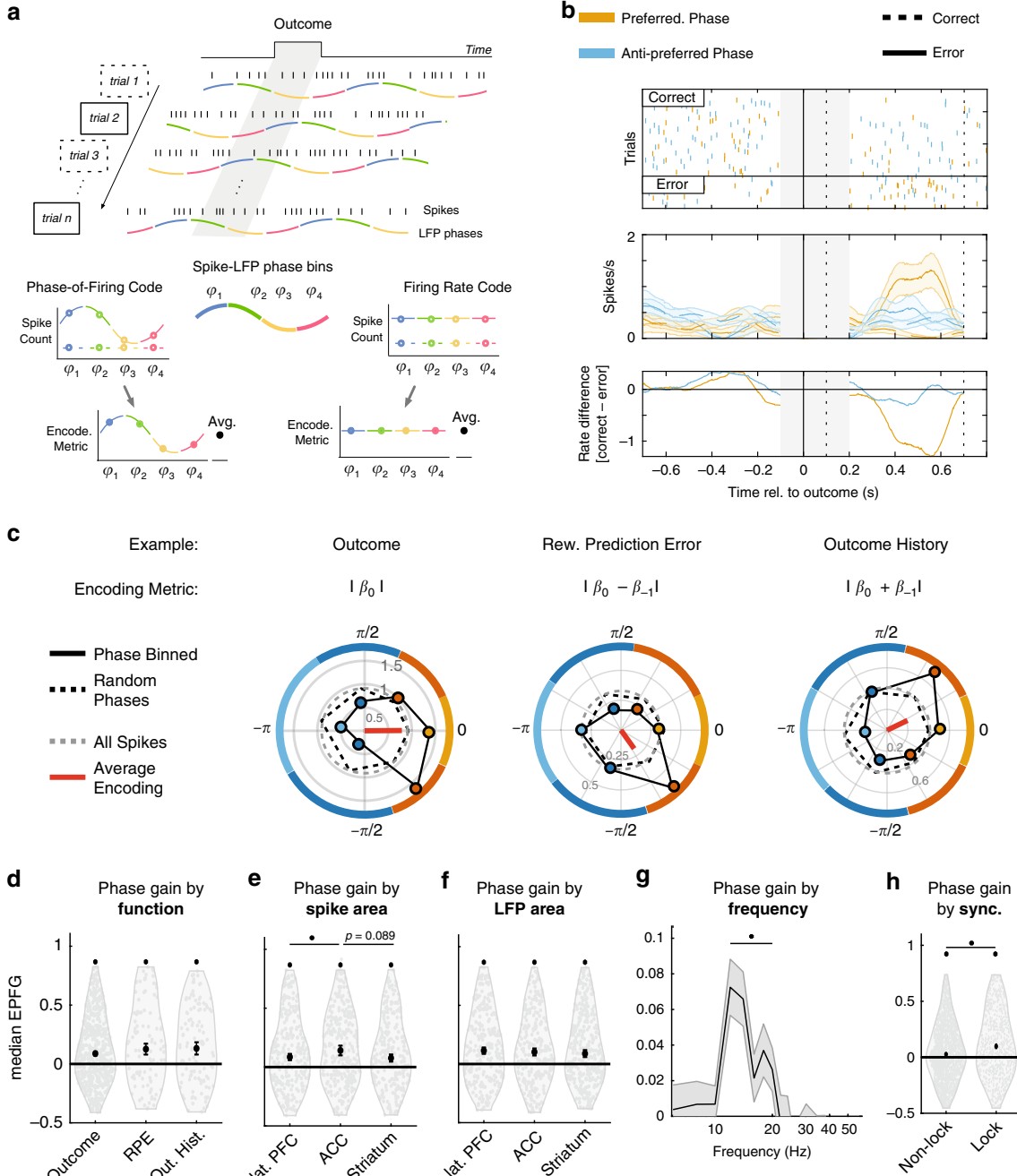

**Fig. 4 Encoded learning signals are modulated by beta phase. a** Phase-dependent encoding analysis. Spikes were segregated by the phase of the LFP, with approximately equal spikes per bin. The encoding metric was extracted for each phase bin. **b** Example cell showing phase-dependent outcome encoding. (top) Raster plot on correct (top) or error (bottom) trials, with firing on the preferred (orange) or anti-preferred (blue) encoding phase. The gray box between [−0.1 0.2]s. was not analyzed as phase was not estimated here. Vertical (dashed) lines depict the period used for GLM analysis. The horizontal black line splits correctly from error trials. (middle) Spike density function for correct (dashed) and error (solid) trials, using spikes from preferred/anti-preferred phases. (bottom) The difference in firing rate between outcomes is greater on preferred, rather than anti-preferred, phases. **c** Example of phase-of-firing encoding for Outcome, RPE, and Outcome History cells. Colored dots reflect the encoding metric for a corresponding phase bin. Colored border lines depict phase bins. Zero radians is the preferred (mean) firing phase. Numbers on concentric circles are the value of the encoding metric. The gray dotted line represents the estimated encoding using all spikes. The black dotted line represents the average across many permutations of spike phases. The red line is the average direction. These cells show stronger encoding near the 0 rad phase, compared to opposite phases. **d** EPFG in each functional cluster. Black circle and vertical bars depict the median and standard error. Violin distributions were trimmed of the top/bottom 5% of data for visualization purposes. All clusters showed some evidence of significant phase gain (Wilcoxon signrank test, $p < 0.05$). (**e**) Same as (**d**) but split by spike area. ACC had a higher EPFG than LPFC or STR. (**f**) Same as (**d**) but split by LFP area. No differences emerge between areas. **g** *EPFG* per frequency for encoding and locking cells ($n = 876$). EPFG was above chance at [10 20] Hz ($p < 0.05$, multiple comparison corrected). **h** EPFG when spikes locked to 10–25 Hz beta phases ($n = 876$), vs. those that did not ($n = 2500$). Non-locking cell showed a significantly less phase gain compared to locking cells (Wilcoxon ranksum test, $p < 0.05$). Stars represent $p < 0.05$.

neuron was in ACC ($Z = 7.5$, $p{\sim}0$), in STR ($Z = 4.98$, $p{\sim}0$), or in LPFC ($Z = 3.86$, $p = 0.0001$) (Fig. 4e), but the EPFG strength differed between areas (Kruskal–Wallis test, $\chi^2 = 7.87$, $p = 0.02$). Neurons in ACC showed significantly larger EPFG than neurons in LPFC ($\chi^2 = 7.66$, $p = 0.0056$) and a trend for larger EPFG than neurons in STR ($\chi^2 = 2.89$, $p = 0.089$). Similarly, spike-LFP pairs with spikes from an ACC neuron were more likely to show individually significant EPFG ($\chi^2$ test, $\chi^2 = 17.7$, $p = 0.0014$; Supplementary Fig. 6). When considering encoding strength on the basis of the LFP site of the spike-LFP pairs, EPFG was above chance in each of the three areas (Fig. 4F; ACC, $Z = 5.02$, p~0; LPFC, $Z = 5.62$, p~0; and STR, $Z = 5.8$, p~0), but did not vary by the LFP area (Kruskal–Wallis test, $\chi^2 = 0.192$, p = 0.91). EPFG differences were more pronounced when selecting for each encoding metric the 25% of spike-LFP pairs with the largest EPFG. This selection revealed stronger EPFG encoding of RPE compared to Outcome ($\chi^2 = 11.3$, $p{\sim}0$) and Outcome History ($\chi^2 = 11.3$, $p = {\sim}0$). It also provided additional confirmation that EPFG was larger for neurons in ACC than in LPFC ($\chi^2 = 10.4$, $p = 0.0013$), with a similar trend for STR ($\chi^2 = 2.41$, $p = 0.12$). Likewise, EPFG did not vary on the basis of LFP area (Kruskal–Wallis, $\chi^2 = 0.192$, $p = 0.91$). These results were similar in each monkey (Supplementary Fig. 4B).

*EPFG* was similar for neurons that showed narrow (N) and broad (B) action potential waveforms that correspond putatively to distinct cell classes with their encoding phase-of-firing gain statistically indistinguishable in the ACC ($N_{\text{N}} = 70$, mean = $-0.026 \pm 0.029$; $N_{\text{B}} = 48$, mean = $-0.0047 \pm 0.06$; Kruskal–Wallis test for equal median, $p = 0.49$), LPFC ($N_{\text{N}} = 85$, mean = $0.11 \pm 0.04$; $N_{\text{B}} = 54$, mean = $0.0057 \pm 0.080$; $p = 0.28$), and STR ($N_{\text{N}} = 37$, mean = $0.014 \pm 0.08$; $N_{\text{B}} = 41$, mean = $0.017 \pm 0.11$; $p = 0.40$) (see "Methods").

We next asked whether the EPFG for RPE encoding distinguishes the rewarded color that animals learned within a reversal block. Previously we showed that the firing rate of subsets of neurons encoded not only a scalar RPE signal but additionally showed stronger RPE signaling for only one or the other color in the task[5]. These color-specific RPE signals can boost the reversal learning because they carry information not only about how much updating should take place (which scalar RPE's signal) but the specific content of what needs to be updated (one or the other color during reversal learning). We quantified this feature-specific RPE encoding by separately testing whether the EPFG is significant when considering only trials when one or the other color was rewarded. We found that of all cell-LFP pairs encoding RPE's, 3% (3/102) showed individually significant EPFG in both conditions (in other words, a non-feature-specific RPE), and ~15% (15/102), showed a significant EPFG only for one of two colors (a feature-specific RPE). The frequency of cell-LFP pairs where the EPFG was significant for neither, both, or only one color condition differed from chance ($\chi^2$ test, $\chi^2 = 109$, $p{\sim}0$). Importantly, feature-specific EPFG was more common than feature non-specific EPFG ($\chi^2 = 6.72$, $p = 0.01$). The proportion of colour-specific EPFG tended to be most prevalent in ACC with ~27% (9/34) of cases, compared to 10% (5/50) in LPFC and 6% (1/18) for STR ($\chi^2$ test, $\chi^2 = 5.26$, $p = 0.07$).

**Robustness of phase-of-firing modulation of encoding**. The EPFG is an effect size measure for how strong firing rate is modulated by LFP phase between conditions. However, it does not take into account the variability of firing rates across trials, leaving open the question of whether such mean firing rate changes may be effectively decoded. To address this question, we performed additional tests at the same beta frequencies at which neurons maximally synchronized. Firstly, we calculated how

much the percent explained deviance varied as a function of phase, which quantifies how well the model fit the data with spikes extracted on individual phase bins. We term this quantity $\text{EPFG}_{D2}$ (see "Methods"). We found that across areas and all spike-LFP pairs with significant encoding, $\text{EPFG}_{D2}$ was significantly larger than chance (Wilcoxon signrank test, $p{\sim}0$). $\text{EPFG}_{D2}$ was significantly above chance for Outcome (Wilcoxon signrank test, $p{\sim}0$) and *RPE* ($p = 0.001$) clusters, but not for spike-LFP pairs with neurons from the Outcome *History* cluster ($p = 0.24$) (Supplementary Fig. 7A).

In a second approach, we tested whether the EPFG is evident even when the statistical testing preserves the within-trial correlation of spike phases. So far, we tested for significance of EPFG by constructing a random distribution that shuffled all spike phases irrespective of the trials in which they occurred. While this preserves the overall degree of synchrony, it destroys any within-trial correlation of spikes. When constructing null distributions by randomly perturbing the phase of spikes on each trial by the same amount, we found an overall significant EPFG of $0.080 \pm 0.018$ (Wilcoxon signrank test, $Z = 7.6$, $p{\sim}0$). As in the other statistics, EPFG was significant for Outcome ($p{\sim}0$), Outcome History ($p = 0.002$), and RPE ($0.039$) (Supplementary Fig. 7B). Similarly, phase-of-firing modulation significantly differed by spike area (Kruskal–Wallis test, $p = 0.032$), with spike-LFP pairs with spikes of neurons in ACC showing higher EPFG than LPFC ($p = 0.012$) and a trend for higher EPFG in ACC than STR ($p = 0.069$) (Supplementary Fig. 7B). Thus, the observed phase gain for the firing rate information is evident even when within-trial autocorrelation is preserved.

In a third approach of analyzing the robustness of the EPFG finding, we considered an alternative normalization of our main encoding metric. The EPFG is a normalized quantity that accounts for the fact that simply fitting a cosine will result in positive amplitudes, implying that a cosine amplitude on its own has an upwards bias. A similar bias is evident in the null distribution of cosine amplitudes. As a consequence, the *EPFG* should be a considered a lower bound on the degree of modulation. This is evident when normalizing the cosine modulation not by the null distribution of the cosine, but by the encoding strength determined using all spikes. With such a normalization, encoding strength is ~$0.61 \pm 0.03$, implying encoding is ~61% stronger on preferred vs anti-preferred phases. Similarly, normalizing the cosine modulation by the over-all firing rate of the cell, we obtained a median EPFG of $0.18 \pm 0.010$, implying that encoding is on average ~18% stronger on preferred rather than anti-preferred phases.

In a final set of analyses, we considered the stability of encoding. Encoding designation was stable across phase bins, with ~90% of spike-LFP pairs exhibiting similar beta coefficient signs across all phase bins, and was not dependent on the number of phase bins used (no correlation of EPFG with [4, 6, 8, and 10] number of bins (Spearman rank correlation, $R = 0.023$, $p = 0.18$).

**Relation of phase-of-firing encoding modulation to the strength and phase of synchronization**. We next tested whether EPFG was specific to the beta frequency band and how the strength of EPFG related to the strength of synchronization. First, we found that EPFG was strongest and significant at the population level in the same beta frequency band that showed the strongest spike-LFP synchronization (Fig. 4g; Wilcoxon signrank test, $p < 0.05$, multiple comparison corrected). Overall, EPFG was most prevalent and significantly larger in spike-LFP pairs that showed significant phase synchronization (Fig. 4h; Kruskal–Wallis test, $\chi^2 = 31.2$, $p{\sim}0$). These results indicate that EPFG was evident when neurons encoded Outcome, RPE, and

Outcome History in their firing rate and when they synchronized at beta-band frequencies.

Second, we tested whether the phase-of-firing modulation of encoding is associated with stronger spike-LFP synchronization in one task condition than in another condition (e.g., in error trials versus correct trials). Such site-specific selectivity of neuronal synchronization has been reported in previous studies (e.g.,[42,46,47].). To test this possibility, we correlated the phase-of-firing encoding with the difference in spike-LFP synchronization (indexed with the PPC) of those two trial conditions that were predicted to have the maximal firing rate difference. For Outcome encoding we calculated the PPC difference for correct versus error trials; for RPE encoding we compared correct trials following error trials versus error trials following correct trials; and for Outcome History encoding we compared correct trials following correct, versus errors following errors. We then correlated the absolute difference in PPC in the beta band between two conditions with the EPFG. We found that the EPFG was uncorrelated with the PPC differences between conditions for neurons encoding RPE (Spearman correlation, $R = 0.083$, $p = 0.36$), or Outcome History ($R = 0.074$, $p = 0.41$). For Outcome encoding cells we found a moderate positive correlation with higher EPFG associated with larger differences in spike-LFP synchronization for correct versus error trials ($R = 0.11$, $p = 0.0067$).

In addition to the strength of synchronization, phase-of-firing modulation of encoding might also become evident as a difference of the preferred phase of synchronization between conditions. To test this possibility, we compared the average phase between conditions for each encoding cluster. We found that for neurons in the Outcome encoding cluster, the mean firing phase in correct and error trials did not differ (mean phase difference $= -0.026 \pm 0.0021$ SE radians, bootstrap randomization test, $p = 0.57$). On the other hand, Outcome History cells significantly synchronized on average at different phases between conditions ($-0.20 \pm 0.0059$ SE radians, $p = 0.011$), with a similar trend for RPE cells ($-0.22 \pm 0.014$ SE radians, $p = 0.059$).

We next asked whether the synchronizing phases that carried information were endogenously generated or whether they were externally triggered by the reward onset. We calculated the EPFG with and without subtracting the reward-onset aligned evoked LFP response (see "Methods"). We found that the EPFG was not different with (median $= 0.096 \pm 0.012$ SE) versus without (median $= 0.10 \pm 0.019$ SE) subtraction of the time-locked, evoked potential, suggesting that the beta oscillation events providing informative phases were endogenously generated (Kruskal–Wallis test, $\chi^2 = 0.03$, $p = 0.86$). In line with this, we found that band-limited power in the beta band was a prominent and sustained component of the LFP after reward onset (Supplementary Fig. 8A) but without a reward-onset locked phase consistency (Supplementary Fig. 8B). We also tested whether LFP power variations or overall firing rate fluctuations influenced the phase-of-firing modulation of encoding. We found that overall the EPFG did not correlate with beta band power variations (Spearman rank correlation, $R = 0.050$, $p = 0.14$), but positively correlated with the overall firing rates of neurons (Spearman rank correlation, $R = 0.13$, $p \sim 0$).

In addition to overall variations of power and firing rates, recent studies have shown that beta-band activity emerges in individual trials as transient bursts that can be linked to behavioral success in working memory and perceptual recognition paradigms[48–50]. To test whether such burst occurrences may underlie the significant EPFG we report so far, we restricted the analysis of the EPFG to those beta band periods that were part of a suprathreshold, oscillatory burst event (see "Methods"). This analysis was performed for spike-LFP pairs when neurons fired

sufficient numbers of spikes (here: ≥30 spikes) per condition. The beta burst rate sharply increased after reward onset, as compared to a pre-reward onset period (see Supplementary Fig. 9A). We found that for spikes occurring within bursts, the median EPFG was $0.067 \pm 0.034$, which was significantly above chance (Supplementary Fig. 9B; $n = 191$; Wilcoxon signrank test, $Z = 2.40$, $p = 0.016$). EPFG for spikes outside bursts was $0.038 \pm 0.017$, which was also above chance (Supplementary Fig. 9B; $n = 769$; $Z = 4.51$, $p \sim 0$). Although encoding was higher inside rather than outside of bursts, this difference was not significant (Kruskal–Wallis test, $\chi^2 = 0.057$, $p = 0.81$).

**Preferred spiking-phase and encoding spiking-phase differ for prediction error.** The previous result suggests that the encoding gain through the phase-of-firing is only weakly or not systematically associated with the strength of spike-LFP phase synchronization. This finding is consistent with a scenario in which the spiking-phase at which neurons maximally synchronize does not always coincide with the spiking-phase at which encoding of task variables is maximal. Indeed, we often observed that the phase with maximal encoding was not at the zero-phase bin, i.e., it deviated from the preferred spike-phase (see examples in Fig. 4c; Supplementary Fig. 5). We tested this scenario by first calculating the preferred spike-phase for each neuron, and then quantifying the phase with maximal encoding relative to that phase. We found that all encoding neurons synchronized on average at similar phases, above what would be expected by chance (Outcome, average phase: $-0.28 \pm 0.0034$ SE radians; Hodges–Ajne test for non-uniformity, $p \sim 0$; RPE. average phase: $0.35 \pm 0.0034$ SE radians, $p = 0.00084$; Outcome History. average phase: $-0.68 \pm 0.0045$ SE radians, $p = 0.0013$) (Fig. 5a). The preferred spike-phase differed between the three encoding classes (Watson–Williams test, $p \sim 0$, $F = 12.8$; each pairwise comparison showed: Watson–Williams test, $F > 7.7$, $p < 0.02$; Fig. 5b).

Next, we quantified for each cluster whether the phases showing maximal encoding were consistent across spike-LFP pairs, because the phase heterogeneity can be informative about possible readout strategies[51,52] (Fig. 5c). To this end, we extracted the phase offset from our cosine fit, which represents the phase at which encoding was maximal relative to the preferred spike-phase. Outcome encoding neurons showed preferred encoding phases that varied across the whole oscillation cycle (average phase: $-0.92 \pm 0.34$ SE radians; Hodges–Ajne test, $p = 0.38$), as did Outcome History neurons (average phase: $1.19 \pm 0.49$ SE radians, $p = 0.66$) (Fig. 5c). In contrast, RPE encoding neurons significantly encoded at similar phase-offsets relative to the neuron's synchronizing phases (average phase: $-2.76 \pm 0.047$ SE radians, $p = 0.0004$, corresponding to 27 ms away from the mean spike phase at a 15 Hz oscillation cycle), which was significantly different than the mean spike phase (Median test, $p = 0.027$). This effect was particularly pronounced for *RPE* cells in ACC (Supplementary Fig. 10), and was consistent across both monkeys (Supplementary Fig. 4C). Qualitatively similar results were obtained when extracting the preferred encoding phases derived from model deviances (Supplementary Fig. 7C). The phase of maximal encoding did not differ with varying number (4, 6, 8, 10) of phase bins used (Circular–Linear correlation, $R = 0.013$, $p = 0.76$). We next compared the relative phases showing maximal encoding between neurons encoding Outcome, RPE, and Outcome History and found that their average, relative encoding phases significantly differed (Watson–Williams test; $F = 83.4$, $p \sim 0$; Fig. 5D). These results show that the preferred spike phase and the encoding phases are typically dissociated from one another, and—for RPE's—were systematically offset from the preferred (mean) spike phase.

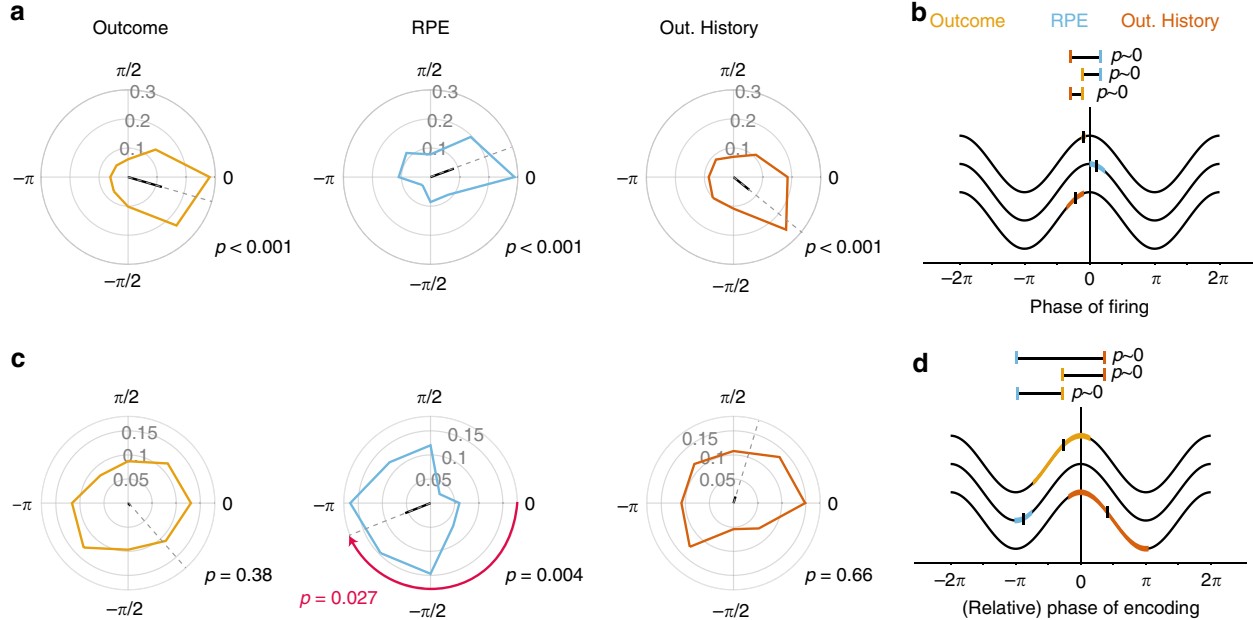

**Fig. 5 Preferred firing and encoding phase are dissociated in RPE cells. a** Proportion of cells with a preferred phase of firing for each encoding cluster. Black lines depict the average phase. Outcome (yellow), RPE (blue) and Outcome History (red) cells showed strong evidence of phase concentration (Hodges–Ajne test, $p < 0.05$). **b** Mean preferred phase of firing for Outcome, RPE, and Outcome History clusters. Colored patches represent the 95% confidence interval about the mean, depicted as a black horizontal line. The distribution of preferred firing phases differed between all encoding clusters (Watson–Williams test, $p < 0.05$). **c** Relative encoding phase (where 0 corresponds to the preferred phase). Outcome and Outcome History cells did not show an encoding-phase preference, whereas RPE cells did (Hodges–Ajne test, $p = 0.0004$). Importantly, for these RPE cells, this phase differed significantly from 0 (the preferred phase of firing; Median test, $p = 0.027$). **d** Similar conventions as in (**b**) but for the encoding phase. The distribution of encoding phases differed between all three clusters.

Given these results, we next tested whether the dissociation of spike- and encoding- phases is not based on possible systematic phase shifts due to differences in the peak oscillation frequencies within the beta band. We validated that this was not the case and found that the three sets of neuronal encoding clusters synchronized on average at the same ~15 Hz center frequency (Kruskal–Wallis, $\chi^2 = 0.95$, $p = 0.62$; Supplementary Fig. 11A), and that they showed maximal phase-of-firing encoding at similar frequencies (also ~15 Hz) (Kruskal–Wallis test, $\chi^2 = 0.39$, $p = 0.82$; Supplementary Fig. 11B). Moreover, the frequency showing strongest spike-LFP synchronization and the frequency showing maximal encoding-phase gain matched closely (median frequency ratio: $1 \pm 0.01$ SE; Supplementary Fig. 11C). This similarity of synchronization and encoding frequency did not differ on the basis of the functional designation (Kruskal–Wallis test, $\chi^2 = 0.047$, $p = 0.98$), nor the area from which the spikes were sampled (Kruskal–Wallis test, $\chi^2 = 0.53$, $p = 0.77$).

## Discussion

Here, we found a significant proportion of neurons whose phase-of-firing in a band-limited beta frequency conveyed significantly more information about three learning variables than their firing rates alone. This encoding-phase gain was evident for spikes generated within the ACC, LPFC and STR of nonhuman primates in a [0.1 0.7] second period of outcome processing during reversal learning performance. Phase-of-firing encoding was most prominent at the 10–25 Hz beta frequency at which spikes synchronized to the local fields across areas. However, the strength of spike-LFP phase synchronization could not necessarily explain the strength of the phase-of-firing encoding. Rather, maximal encoding occurred for many neurons at phases away from the preferred spiking phase. The dissociation of spiking and

encoding phases was particularly prominent for information about the RPE.

Taken together, these results show population-level information multiplexing of learning variables at segregated phases of a beta oscillation across synchronized medial and lateral fronto-striatal loops. These findings suggest that spike-LFP oscillation phases are carriers of information, above and beyond that of a phase-blind firing rate code. The gain of information through the phase of firing provides an intriguing dynamic code that could link principles of efficient neuronal information transmission with the demands of representing multiple types of information in the same dynamical neural system.

We found that three critical variables needed for adjusting behavior are represented in partly segregated neuronal populations not only in their firing rates, but in phase-specific firing at a beta frequency that is shared among ACC, LPFC and STR. This finding suggests that the beta frequency could serve as an important conduit for the fast distribution of learning-related information within fronto-striatal networks[19,20]. Prior studies have shown that ACC, LPFC and STR causally contribute to fast learning of object values. With lesioned ACC, rhesus monkeys fail to use outcome history for updating values and show perseverative behaviors[53]. Without LPFC, rhesus monkeys fail to recognize when a previously irrelevant object becomes relevant as if they fail to calculate *RPE*'s needed for updating their attentional set[54]. When the anterior STR is lesioned, nonhuman primates tend to stick to previously learned behavior and show a lack of sensitivity to reward outcomes[4,35]. These behavioral lesion effects are consistent with the important role of each of these brain areas to track the history of recent outcomes, registering newly encountered (current) outcomes, and calculating the unexpectedness of experienced outcomes (prediction error). Consequently, our finding of segregated neuronal ensembles encoding Outcome,

Prediction Error and Outcome History complements a large literature that documents how these variables are represented in the firing of neurons across fronto-striatal areas.

What has been left unanswered, however, is how this firing rate information about multiple variables emerges at similar times and similar proportions across areas. Prior studies suggest that firing rate correlations between brain areas are relatively weak and poor candidates for veridical information transfer[5,8,26], while temporally aligning the spike output of many neurons to the phases of precisely timed, synchronized packets are a theoretically, particularly powerful means in affecting postsynaptic neuronal populations[24,26,27,55,56]. Our findings support this notion of a temporal code using synchronized oscillations by showing that those neurons that carry critical information in their firing rates also tend to synchronize long-range between ACC, LPFC, and STR at a shared 10–25 Hz beta frequency. This beta frequency is thus a candidate means for enhancing distributed information transfer, because spike output of many neurons is concentrated at the same phase and thus activate postsynaptic membranes at similar times. This scenario of beta rhythmic information exchange within fronto-striatal networks is supported by previous nonhuman primate studies that demonstrated 10–25 Hz beta rhythmic synchronization during active task processing states between ACC and LPFC[12,13,57], between PFC and STR[15], between ACC and FEF[10], between LPFC and FEF[46,47], and between LPFC or FEF with posterior parietal cortex[46,47,58–60]. Each of these studies has shown short-lived rhythmic long-range synchronization between distant brain areas during cognitive tasks at a ~15 Hz frequency, similar to studies in humans (e.g.,[61]). Our findings critically complement these studies by revealing that 10–25 Hz spike-LFP synchronization is prevalent not only during cognitive processing, but also during the processing of outcomes after attention has been deployed and choices have been made. During this post-choice outcome processing, fronto-striatal circuits are likely to adjust their synaptic connection strength to minimize future prediction errors and improve performance[4,5,62]. Our results suggest that this updating utilizes beta rhythmic activity fluctuations during the post-choice outcome processing period.

Our finding that spiking output carries separable types of information at different phases of the same oscillation frequency has potentially far-reaching implications. By finding that Outcome, Prediction Error and Outcome History were encoded at separate phases, the population of neurons effectively multiplexes independent information streams at different phases of beta synchronized firing. This stands in contrast to prior studies reporting that long-range beta rhythmic synchronization between LPFC, ACC or STR in the primate encoded relevant task variables via the strength of beta synchrony[10,15,47,58,63]. For example, some prefrontal cortex neurons synchronize stronger at beta to posterior parietal areas when subjects choose one visual category over another[46], or when they maintain one object over another in working memory[47]. These findings are broadly consistent with a communication-through-coherence schema where upstream senders are more coherent with downstream readers when they successfully compete for representation[24,64,65]. Yet it has remained unclear how such a scheme may operate when multiple items must be multiplexed and transmitted in the same recurrent network[7,23,28,66–68]. Computationally, the multiplexing and the efficient transmission of information can operate in tandem when the temporal organization of activity is exploited at the sending and receiving site[8,26,27,69]. Consequently, selective synchronization between distal sites could be leveraged to enhance transmission selectivity, whereas temporally segregated information streams could enhance transmission capacity[70]. Our results resonate with this view by showing that neurons that synchronize long-range at one oscillation phase carries information of any of three learning variables at phases systematically offset from the synchronizing phase.

By finding evidence for such a temporal multiplexing in the beta frequency band, we critically extend previous reports of phase encoding of information for object features, object identities, and object categories at theta, alpha and gamma frequencies[45,67,71–73]. In our study, the beta phase enhanced encoding applied to three complex learning variables that were needed to succeed in the behavioral learning task. In particular, the presence of reward prediction error information provides a critical teaching signal that indicates how much synaptic connections should change to represent future value expectations more accurately[5,62]. Our results suggest that this updating can utilize spike-timing-dependent plasticity mechanisms that are tuned to firing phases ~27 ms away from the preferred synchronization phase in the beta frequency band. How such a temporal organization in the beta band is used in the larger fronto-striatal network will be an important question for future studies.

A caveat in interpreting the phase-modulated coding we report is that it is consistent with multiple coding schemes beyond a phase-based multiplexing[74]. For example, spiking activity may be phase-synchronized in one condition but not another, or alternatively, conditions may be encoded on separate phases. Our results provide support for both coding schemes. Outcome cells resemble coding via an asynchronous code; that is to say, spike-LPF phase synchronization is higher in one condition than another, with no evidence of phase differences between conditions. On the other hand, RPE and Outcome History cells show evidence of phase-separation coding. These cells showed no significant difference in PPC between conditions but did show a (near significant) trend towards firing on different phases. These suggestions depend on a proper estimation of phase, which can be influenced by the level of background noise[75] and the degree of synchrony of individual cells within a population[76]. However, we believe this would not affect the main conclusions of our study, as we observed (1) significant increases in encoding-phase gain both when oscillatory bursts were prominent or not, and (2) significant phase encoding gain when controlling for outcome induced activity.

So far, evidence in humans and rodents suggested that processes linked to beta frequency activity during the evaluation of outcomes support the detection of errors and the updating of erroneous internal predictions[17,77]. In fact, there have been conflicting views on whether beta oscillations related to outcome signals are more likely to reflect a weighted integration of recent outcomes, or the unexpectedness of the current, observed outcome relative to recent outcomes[17,21]. Our findings reconcile these viewpoints by documenting that encoding of Outcome History weights and of Prediction Errors coexist in the same circuit at the same oscillation frequency in phase-dependent firing of single neurons.

We found that the beta phase allowing maximal encoding of Prediction Errors was offset ~27 ms on average from the phase at which most spikes synchronized to the local fields. Such a dissociation of mean spike-phase and encoding-phase has been reported previously for the beta frequency band in parietal cortex, where maximal information of joint saccadic and joystick choice directions were best predicted by spike counts at ~50 degrees away from the preferred beta spike phase[44]. Such phase offsets underlying maximal encoding in parietal cortex as well as in ACC, LPFC, and STR in our study provide constraints on the possible circuit mechanisms that permit temporal segregation of inputs streams through phase-specific oscillatory dynamics[7]. One possible circuit mechanism that implements and utilizes

multiplexed information streams through phase-specific firing has been described and computationally modeled specifically for the low 10–20 Hz frequency range[23,78]. This work suggests that distinct sets of pyramidal neurons can encode distinct input streams in their firing phases at 10–20 Hz beta activity when these inputs streams arrive with a phase offset to each other, e.g., when they arrive sequentially in time. According to this schema, a first input stream activates pyramidal neurons in deep cortical layers that feed information to superficial layers whose interlaminar inhibitory connections closes an interlaminar reverberant loop of activity. This interlaminar ensemble follows a beta activity rhythm due to cell specific dynamics that maintains the beta-phasic firing of active neurons[23,43]. When a second input stream activates another set of pyramidal cells within the same beta rhythmic neural population, the input timing of that second stream was maintained at a different phase than the phase of the first activated ensemble[23]. The parallel coding of information at a common beta rhythm in these models provides a qualitative proof of concept about phase-specific encoding of multiple types of inputs in larger beta rhythmic ensembles, and suggests a possible mechanistic realization of enhanced encoding by the phase of firing in the beta band[23]. Moreover, these models[23,43] also suggest possible reasons why encoding phases and the average, preferred spiking phases can differ. In our study RPE encoding was maximal for spikes that occurred 27 ms away from the preferred beta phase at which most spikes of the neurons were elicited. In the context of these models, a phase offset could indicate that RPE's are part of an input stream that is arriving already with a delay to the major beta rhythmic input stream that this neuron sees. For example, input carrying prediction error information might arrive from the ventral tegmental area while the dominant beta rhythmic firing (that determines the mean phase) might be based on local cortical mechanisms coupled to other cortical areas. Consistent with such a scenario, a prior rodent study[79] has shown that the phase of phase-synchronous prefrontal cortex neurons shifted with the learning of new reward locations, consistent with a dopaminergic influence form the ventral tegmental area on the phase of spike-LFP synchrony. Alternatively, a 27 ms phase offset for encoding prediction error information might have a local origin, with the delay reflecting the computation of the error in prediction based on input that carries the prediction itself. This scenario gains plausibility when considering that a prediction error reflects a transformation of two signals, i.e., it is the difference of the expected value and the received outcome. This transformation will take time. In the temporal domain, this delay is likely reflected in a latency difference with prediction error signals emerging typically after outcomes are processed (which we found, Supplementary Fig. 2G, H). In a recurrent circuit, this delay in computing an error might additionally be reflected in a phase offset. According to this view, the 27 ms offset in maximal encoding of RPE indicates a local transformation of two input streams (predicted value and outcome) into their difference (the error in value prediction). Future work needs to specify whether these scenarios are realized by beta rhythmically firing ensembles of neurons and how long-lasting and robust the encoding with phase-specific firing is with regard to the overall firing rates and firing variability of individual neurons during active brain states.

In summary, we have documented that learning variables are better encoded when taking into account the phase of firing of neurons that synchronize long-range across primate fronto-striatal circuits. These neurons that showed a phase encoding gain of their firing also carried information in overall firing rate modulations which clarifies that an asynchronous rate code and a synchronous temporal code coexist in the same circuit[8]. By exploiting the temporal structure endowed in long-range neuronal synchronization our findings suggest how neuronal

assemblies in one brain area could be read out from neural assemblies in distally connected brain areas[29]. This phase-of-firing schema entails key features required from a versatile neural code including the efficient neural transmission and the effective representation of variables needed for adaptive goal-directed behavior[80].

## Methods

**Experimental animals.** Data were collected from two adult, 9 and 7-year-old, male rhesus monkeys (*Macaca mulatta*) following procedures described in ref. [5]. All animal care and experimental protocols were approved by the York University Council on Animal Care and were in accordance with the Canadian Council on Animal Care guidelines.

**Behavioral paradigm.** Monkeys performed a feature-based reversal-learning task that required covert attention to one of two stimuli based on the reward associated with the color of the stimuli. Which stimulus color was rewarded remained identical for ≥30 trials and reversed without explicit cue. The reward reversal required monkeys to utilize trial outcomes to adjust to the new color-reward rule. Details of the task have been described before[5]. Each trial started when subjects foveated a central cue. After 0.5–0.9 s, two black and white gratings appeared. After another 0.4 s, the stimuli either began to move within their aperture in opposite directions (up-/downwards) or were colored with opposite colors (red/green or blue/yellow). After another 0.5–0.9 s, they gained the color when the first feature was motion, or they gained motion when the first feature had been color. After 0.4–0.1 s, the stimuli could transiently dim. The dimming occurred either in both stimuli simultaneously, or separated in time by 0.55 s. Dimming represented the go-cue to make a saccade in the direction of the motion when it occurred in the stimulus with the reward associated color. The dimming acted as a no-go cue when it occurred in the stimulus with the non-rewarded color. A saccadic response was only rewarded when it was made in the direction of motion of the stimulus with the rewarded color. Motion direction and location of the individual colors were randomized within a block. Thus, the only feature predictive of reward within a block was color. Color-reward associations were constant for a minimum of 30 trials. Block changes occurred when 90% performance was reached over the last 12 trials, or 100 trials were completed without reaching criterion. The block change was uncued. Rewards were deterministic.

**Electrophysiology.** Extra-cellular recordings were made with 1–12 tungsten electrodes (impedance 1.2–2.2 MOhm, FHC, Bowdoinham, ME) in ACC (ACC; area 24), prefrontal cortex (LPFC; area 46, 8, 8a), or anterior STR (STR; caudate nucleus (CD), and ventral striatum (VS)) through a rectangular recording chambers (20 by 25 mm) implanted over the right hemisphere (Supplementary Fig. 1). Electrodes were lowered daily through guide tubes using software-controlled precision micro-drives (NAN Instruments Ltd., Israel and Neuronitek, Ontario, Canada). Data amplification, filtering, and acquisition were done with a multi-channel acquisition system (Neuralynx). Spiking activity was obtained following a 300–8000 Hz passband filter and further amplification and digitization at 40 kHz sampling rate. Sorting and isolation of single unit activity was performed offline with Plexon Offline Sorter, based on analysis of the first two principal components of the spike waveforms. Experiments were performed in a custom-made sound attenuating isolation chamber. Monkeys sat in a custom-made primate chair viewing visual stimuli on a computer monitor running with a 60 Hz refresh rate. Eye positions were monitored using a video-based eye-tracking system (EyeLink, SRS Systems) calibrated prior to each experiment to a nine-point fixation pattern. Eye fixation was controlled within a 1.4°–2.0° radius window. During the experiments, stimulus presentation, monitored eye positions, and reward delivery were controlled via MonkeyLogic. The liquid reward was delivered by a custom-made, air-compression controlled, and mechanical valve system. Recording locations were aligned and plotted onto representative atlas slices[81].

**Data analysis.** The analysis was performed with custom Matlab code (Matlab 2019a), using functions from the fieldtrip toolbox. For Elastic-net regression, the *glmnet* package in R was used[82]. Only correct and error responses were analyzed. Error responses included those where the responses were made to the incorrect target, or in the incorrect response window. We included all trials from learned blocks, with a minimum of two blocks, unless otherwise indicated. The trial immediately following a reversal event was not included in analysis. Learned blocks were defined as ones where animals reached 90% correct responses within the last 10 trials within the block. Standard errors of the median were estimated via bootstrapping (200 repetitions, unless otherwise indicated).

Data analyses proceeded through multiple steps. First, we quantified how outcomes (reward/no-reward for correct/error outcomes) affected monkeys' choices. After showing that outcomes are integrated over recent trials, we next asked how this is reflected in the firing rate activity of individual neurons during a post-outcome period using a penalized GLM. We used a data-driven clustering approach to assign functional labels to cells exhibiting similar sensitivities to experienced outcomes in their rate. On the basis of these functional labels, we

extracted a corresponding encoding metric for neurons in each functional cluster. We then analyzed how the encoding metrics depend on time, or the phase of oscillatory activity in the LFP. For the latter analysis, we used standard spectral decomposition techniques and spike-phase consistency measures to characterize how spikes and phases between distal electrodes are related. We quantified and compared differences in phase-dependent encoding in terms of (1) the degree of phase-dependent modulation of encoding, and (2) the phase at which encoding is maximal.

**Behavioral analysis**. To determine the timescale over which past outcomes are integrated, we used a binomial GLM:

$$Y = \sum_i^5 \beta_{t-i} X_{t-i}, \tag{1}$$

where $Y$ was the current outcome, $B_{t-i}$ is the influence of outcome $X_{t-i}$ on trial $t-i$. The outcome for trial $t-5$ was defined as a nuisance variable that accounted for all responses occurring over very long time-scales (similar to ref. [33]).

**Rate encoding of outcome history**. To test how individual units integrated outcome history, we used a Poisson GLM:

$$\log(\lambda) = \sum_i^5 \beta_{t-i} X_{t-i}, \tag{2}$$

where $\lambda$ was the conditional intensity (spike count), $B_{t-i}$ is the influence of outcome $X_{t-i}$ on trial $t-i$. Firing counts on each trial were determined in a [0.1 0.7]s window after outcome onset[5]. Neurons were included in the analysis if they were isolated for more than 25 (learned) trials across at least two blocks, and if they showed an overall firing rate of >1 Hz. With these criteria, we analyzed a total of 1460 neurons, with an average of 230.56 ± 3.44 trials and 5.75 ± 0.082 blocks.

To mitigate issues of multi-collinearity, and extract only the most predictive regressors, we employed elastic-net regularization using the R package *glmnet*[82]. This procedure shrinks small coefficients to zero, and smoothly interpolates between ridge and lasso regularization by controlling a parameter alpha (with alpha = 0 corresponding to ridge regression, and alpha = 1 to lasso regression)[82]. We used an alpha of 0.95, which tends to select only one regressor in the presence of collinearity (as in pure lasso regression[83]), while at the same time avoiding issues with degeneracy if correlations among regressors are particularly strong[82]. The optimal value of the shrinkage parameter (lambda) was the minimum as selected by 10-fold cross validation. To assess model stability and extract significant fits, we used a bootstrap approach, whereby trials were sampled with replacement 1000 times and the procedure was rerun. As the LASSO shrinks non-valuable predictors to zero, a model fit was said to be significant if at least one relevant regressor (outcome $t-4$ to $t-0$) was non-zero more than 95% of the time.

**Functional clustering based on neural encoding**. Our ultimate goal is to describe how encoding varies as a function of phase (and time). However, encoding cells showed variability in how they responded to experienced outcome (e.g., Fig. 2; Supplementary Fig. 2D). Thus, in order to properly evaluate changes in encoding in time and phase, we must first define populations of cells that encode similar types of information. To determine the putative function of significantly encoding units, we used a clustering approach via bootstrapped K-means. We clustered cells on the basis of their mean beta weights as determined by the penalized regression model (see above). As a preprocessing step, for units where the current outcome was negatively encoded (i.e., encodes errors), we flipped the sign of every coefficient in that model. This has the effect of erasing the directionality of any functional association, and thus collapses neurons with similar functions (for example, Error or Correct encoding units become Outcome encoding units). Cells were independently clustered for each area.

We clustered cells on the basis of their clustering stability[84]. We opted for this method because k-means clustering can be sensitive to initial conditions[85]. This involved three steps: (1) choosing the optimal number of clusters $Nc$, (2) measuring clustering stability, and (3) performing the final clustering. For steps 1–2, we used k-mean clustering with a cosine distance metric, which is insensitive to the magnitude of the vector and is instead concerned with the direction, unlike, for example, the Euclidian distance. In other words, we clustered based on the relative pattern of beta weights of each cell, irrespective of differences in magnitude between cells.

To determine the optimal number of clusters, we extracted the Silhouette metric over many bootstrap iterations. In brief, cells were sampled with replacement 1000 times and for each iteration, the optimal number of clusters was extracted where the silhouette was maximal. The overall optimal number of clusters $Nc$ was the mode over all bootstrap iterations.

Next, we assessed the clustering stability of pairs of cells. To do so, we built a similarity matrix $S$ via a bootstrap approach, where similarity was defined as the proportion of times that pairs of cells were clustered together. First, we resampled with replacement individual cells. Next, we ran K-means with cosine distance and $Nc$ clusters. For units that were clustered together, their respective cell in the

similarity matrix was incremented by one. Because bootstrapping could sample the same units twice, these pairs were ignored. Bootstrapping was run 100,000 times.

To compute the final cluster assignment, we first formed a dissimilarity matrix $D = 1 - S$, before performing agglomerative clustering with Euclidian distance and $Nc$ clusters.

**Metric for outcome, outcome history, and prediction error**. We quantified the degree of encoding of Outcome ($E_{\text{outcome}}$), Outcome History ($E_{\text{history}}$), and Reward Prediction Error ($E_{\text{RPE}}$) on the basis of the GLM weights for trials $-1$ and $0$:

$$E_{\text{outcome}} = \text{abs}(B_0), \tag{3}$$

$$E_{\text{history}} = \text{abs}(B_{-1} + B_0), \tag{4}$$

$$E_{\text{RPE}} = \text{abs}(B_0 - B_{-1}) \tag{5}$$

We refer to these generically as Encoding Metrics.

**Latency analysis**. To determine the latency of encoding for each functional cluster, we performed a time-resolved analysis (Supplementary Fig. 2G, H). On the basis of our previous results showing that the outcome on trial 0 and $-1$ were most predictive (Fig. 2), we used a simpler GLM of just the current and previous outcome. For the response variable, we calculated the spike density using a sliding Gaussian window, with a 200 ms window and 50 ms standard deviation. We performed this analysis [$-0.4$ 0.7] around outcome onset. We thus obtained a time-resolved estimate of encoding.

To determine the latency of significant encoding, we looked at time points in the post-outcome period that were significantly different from the pre-outcome period. We thus determined, for each cell, when encoding exceed a threshold criterion in a time-of-interest. First, we z-score normalized each individual cell's encoding metric to the pre-outcome period ([$-0.4$ 0] s). Next, we asked, for each time point, whether the population response was significantly different from zero via a Wilcoxon signrank test. We then extracted the largest cluster mass of contiguous significant time points (at an alpha = 0.05, e.g.,[86]) to find a time-of-interest. Finally, we extracted, for each individual cell, the time point where the area under the curve of the encoding metric in this time-of-interest reached 10% of the total. Thus, we obtain for each encoding cluster a distribution of latencies of when they started to show significant encoding of Outcome, Outcome History, or Prediction Error.

**Spectral decomposition and spike-LFP phase synchronization**. To determine how encoding varied as a function of phase, we extracted the estimate of phase at the time of spikes, for frequencies from 6 to 60 Hz. We first characterized the degree of spike-phase synchronization, described below. We focused spike-phase analysis on pairs of distally recorded sites, thus obviating any concerns of spike energy bleeding into the LFP[87]. For frequencies from 6 to 30 Hz, the resolution was 1 Hz, and above that it was 2 Hz. For every frequency $F$, we determined the spike-LFP phase by extracting an LFP segment centered on the spike of length $5/F$ (i.e., 5 cycles), as is standard to balance temporal and spectral resolution. Spectral decomposition was done via an FFT after applying a Hanning taper. This procedure was applied separately to the pre-outcome period [$-1$ 0]s, and the post-outcome period [0.1 1]s.

The strength of spike-LFP synchronization was quantified using the pairwise-phase consistency (PPC), which is unbiased by the number of spikes[41]. The PPC is quantified on the basis of pairwise differences between spike-phases. If spikes tend to fire on specific phases, phase difference will be concentrated, and thus the PPC will take on a high value, whereas if spikes are distributed randomly relative to the LFP phase, phase differences will be random and the PPC will tend towards zero. The PPC effect size was determined as previously reported[12,13]

$$\text{Effect size} = \frac{1 + 2*\text{sqrt(PPC)}}{1 - 2*\text{sqrt(PPC)}}. \tag{6}$$

This effect size can be interpreted as the relative increase in spike rate at the cell's preferred (mean) phase over its anti-preferred (opposite) firing phase. For example, a PPC value of 0.01 corresponds to a 1.5 times greater spike rate at the preferred phase.

We determined the frequency at which spike-LFP phase synchronization was significant by determining peaks in the PPC spectrum. A cell was said to synchronize to a particular frequency if the following criteria were met: (1) Peaks had to be above a threshold of 0.005, (2) show a minimum prominence of 0.005, and (3) show significant Rayleigh test (i.e., phase concentration).

To test for inter-areal differences in spike-beta synchronization, we extracted the maximal significant/prominent PPC peak in the [10 25] Hz band that showed significant encoding. For those encoding cells that did not show significant PPC peaks, we extracted the frequency of the maximal PPC in this band instead. We tested for differences in synchronization strength using a one-way ANOVA, and report on pairwise comparisons after multiple comparison correction.

**Phase-of-firing dependent encoding of outcome, outcome history, and prediction error**. To determine if spikes falling on certain phases of the LFP were more informative, we re-ran the (reduced) GLM on phase-binned spikes, using only the previous and current outcomes (see "Latency analysis" above).

We first aligned all spike-triggered-phases to the circular mean of their distribution. Phases were extracted from the frequency of the corresponding maximal peak in the [10 25] Hz band in the PPC. However, if spikes are phase locked to an LFP, the firing rate around the preferred phase will naturally be higher. Thus, we used non-equal bin sizes, adjusting the bin limits such that they had the same spike count. On average the spike-count equalized bin has a spike rate of 1.85 Hz, range: 1.80–1.86 Hz across bins). Phase bins with equalized spike counts were on average 7.5% larger for the bin around the non-preferred phase (spanning ~21% of the full cycle) than for the bin around the preferred phase (spanning ~13.5% of the full cycle).

We then re-ran the GLM analysis on spikes falling within a particular bin and computed the encoding metrics as described previously. To aid in comparison, we also fit the model using randomly permuted phases (thus preserving the over-all rate response structure). We ignored spike-LFP pairs where the GLM could not converge to a solution and threw a warning, or where the beta coefficients were above 20 (however, relaxing or tightening this constraint did not qualitatively change the results).

To determine the phase and degree of phase-dependent encoding, we fit a cosine function to the phase-binned encoding values (illustrated in Fig. 4a)[45,67]. One encoding value was selected for each spike-LFP pair, on the basis of the cluster assignment of the spiking neuron. From this fit, we obtain three values: $T$ (phase offset, or phase of cosine maxima), $A$ (amplitude), and $M$ (overall mean, or offset). The value $T$ is thus the phase at which encoding is maximal, relative to the preferred firing phase. To compare the strength of encoding across functional clusters, we computed the empirical phase-of-firing gain:

$$\text{PFG}_E = 2 * \frac{A}{M}. \tag{7}$$

This quantity represents the difference in encoding between the peak and trough relative to the overall encoding strength. A $\text{PFG}_E$ value of 0 implies that phase-of-firing adds no information (corresponding to a pure rate code), whereas $\text{PFG} = 1$ means that encoding between the peak and trough is 100% stronger compared to the overall encoding strength. To determine if phase significantly added information above that of a phase-blind rate code, we opted for a randomization approach. For each cell, we first permuted the phase label of each spike, re-ran the GLM, re-fit a cosine and extracted the encoding phase-of-firing gain. This procedure was repeated 50 times, from which we obtain a distribution $\text{PFG}_R$ of randomized encoding gains. For this procedure, because phase labels were permuted, the distribution of phases remains the same, and thus the bin-widths need not be re-calculated. We report on the "excess" PFG, defined as the difference between empirical and the median of the randomized phase-of-firing gain, which we refer to in the manuscript at the Encoding Phase-of-Firing Gain (EPFG):

$$\text{EPFG} = \text{PFG}_E - \text{median}(\text{PFG}_R). \tag{8}$$

A positive value implies that encoding is modulated by phase above what would be expected by chance.

To assess whether individual units showed significant encoding, we compared $\text{PFG}_E$ against the null distribution $\text{PFG}_R$. Units were deemed significant at an alpha level of 0.05.

The procedure described above destroys any within-trial correlation between spike phases. Thus, in a related analysis, we determined $\text{PFG}_R$ by adding a random phase in the range [0 2pi] to all spikes within a single trial, thus preserving their correlation structure. In this case, the phase bin widths were re-calculated for every randomization.

The EPFG effectively quantifies the difference in mean firing rates between conditions, as a function of LFP phase. However, this does not necessarily imply that the information is easily decodable by other brain circuits. To address this question, we asked how much variance can be explained by the model fit to data in each phase bin. To this end, for each fit on each phase bin, we extracted the percent deviance explained (analogous to the ANOVA percent variance explained but modified for a Poisson GLM). The percent deviance explained $D^2$ was calculated as[88]:

$$D^2 = 1 - \left( \frac{\text{Residual Deviance}}{\text{Null Deviance}} \right). \tag{9}$$

The deviance for a Poisson distribution is defined as:

$$\text{Deviance} = 2 * \sum_i^n Y_i * \log\left( \frac{Y_i}{\lambda_i} \right) - (Y_i - \lambda_i), \tag{10}$$

where $Y_i$ is the observed spike count on trial $i$, and $\lambda_i$ is the predicted spike count. We then determined how $D^2$ varied as a function of phase using the same procedure as described above; namely, we fitted a cosine to the $D^2$ of each phase bin, extracted the amplitude and phase, and compared it to a null distribution where phases have been permuted. We call this quantity the Encoding Phase-of-Firing Gain ($D^2$), or $\text{EPFG}_{D2}$.

We tested the stability of encoding across phase bins for each neuron (with significant rate encoding) by determining the sign of the encoding metric (i.e.,

before taking the absolute value). We found that for the vast majority of cell-LFP pairs (~90%), the sign of the encoding metric was the same for all 6/6 phase bins as for the full model.

To test the frequency specificity of the EPFG, we extended the above analysis to the larger 6–60 Hz frequency range (Fig. 4g). We statistically tested the EPFG across frequencies using the Wilcoxon signrank test. To correct for multiple comparisons, we used a cluster-based permutation approach[89]. First, we determined the largest cluster mass of contiguous significant samples ($p < 0.05$). Next, we shuffled empirical and randomized $\text{PFG}_E$ and $\text{PFG}_R$ across cell-LFP pairs to determine a randomized $\text{EPFG}_R$ and re-calculated the largest cluster mass. We performed this procedure 200 times. Significant clusters were those whose mass exceeded that of the randomized distribution.

We also tested the degree to which our results may be influenced by cue-aligned activity. To this end, we first obtained the average evoked potential for each LFP channel and subtracted this component from individual trials. We then performed all steps of the analysis again to compare the original EPFG with the EPFG free from potential cue-aligned biases.

To test whether the preferred firing phase or relative phase with maximal encoding was concentrated above what would be expected by chance, we used the circular Hodges–Ajne test (Fig. 5). To determine whether the phase showing maximal encoding differed from the preferred firing phase in each functional encoding cluster, we performed the Median test to test if the phase differed from zero[90] (Fig. 5b).

We tested how the strength of phase synchronization related to the strength of phase-of-firing encoding by performing two analysis. First, we compared encoding in cells that showed significant spike-phase synchronization to those that did not. For non-synchronizing cells, we selected the center frequency with the maximal PPC in the [10 25] Hz range, and computed the EPFG at this frequency. We compared EPFG between locking and non-locking populations using the Kruskal–Wallis test (Fig. 4h). Second, we asked whether spike-phase synchrony in different trial conditions contained similar information to that of the phase-of-firing. To this end, for each encoding cell, we compared trials that were predicted to have the maximal firing rate differences. For Outcome encoding, we compared correct versus error trials. For Reward Prediction Error encoding we compared correct trials following error versus following error trials following correct. For Outcome History cells, this was errors followed by errors versus correct outcomes followed by correct. We took the absolute difference of the PPC between the two conditions and correlated it with the EPFG of the respective cell using the Spearman rank correlation. In a similar vein, we also tested whether the mean phase differed between the conditions outlined above. After extracting the mean phase per condition for each cell-LFP pair, we performed a bootstrap test to test if the difference in phase between conditions differed from zero[90].

We also tested whether phase gain depended on the number of bins used to fit the cosine function. We performed the analysis for 4, 6, 8, and 10 bins. We used Spearman rank correlation to determine if EPFG was related to the number of bins, and circular–linear correlation to associate the phase of maximal encoding with the number of bins[90].

We tested for the presence of feature-specific phase-of-firing encoding for those cells clustered as RPE encoding. We calculated the $\text{PFG}_E$ for each cell-LFP pair twice, using only trials from blocks where either color 1 or 2 was rewarded. We analyzed pairs with a minimum of 30 trials, and where the $\text{PFG}_E$ was well-defined for both colors. A total of 102 pairs were thus selected. The average number of trials for color 1 was $136 \pm 5.4$, from an average of $3.15 \pm 0.13$ blocks. Color 2 analysis used $129 \pm 4.85$ trials from $3.07 \pm 0.12$ blocks. We then asked, for each color, whether the $\text{PFG}_E$ was above chance (described above). $\text{PFG}_E$ could be significant for neither color, one-color (defined as feature-specific encoding), or both colors (non-feature-specific encoding). We tested whether the relative frequencies of non-encoding, feature-specific encoding, and non-feature specific encoding differed by chance using a $\chi^2$ test. We used the same test to determine if the proportion of feature-specific encoding differed between areas.

**Cell-type classification and analysis**. To determine if phase-modulated encoding of information differed based on cell type, we focused the following analysis on highly isolated single units that showed encoding of learning-relevant variables and significant, prominent spike-beta locking. Detailed information is provided in ref. [5]. In brief, to distinguish putative interneurons (narrow-spiking) and putative pyramidal cells (broad-spiking) in LPFC and ACC, we analyzed the peak-to-trough duration and the time for repolarization for each neuron. After applying principal component analysis (PCA) using both measures, we used the first principal to discriminate between narrow and broad-spiking cells. This allowed for better discrimination than using either measure alone. We confirmed that a two-Gaussian model fit the data better than a one-Gaussian model using the Akaike and Bayesian Information Criterion (AIC, BIC). We then used the two-Gaussian model to define narrow and broad-spiking populations.

A similar analysis was applied to striatal units to distinguish putative interneurons from medium spiny projection neurons (MSN). Here, we use the peak-width and Initial Slope of Valley Decay (ISVD)[5]:

$$\text{ISVD} = \frac{V_t - V_{0.26}}{A_{\text{PT}}}, \tag{11}$$

where $V_T$ is the most negative value (trough) of the spike waveform, $V_{0.26}$ is the voltage at 0.26 ms after $V_T$, and $A_{PT}$ is the peak-to-trough amplitude. After PCA and two-Gaussian modeling (as described above), we defined two cut-off points. The first cutoff was the point at which the likelihood of narrow-spiking cells was three times larger than the likelihood of broad-spiking cells, and vice-versa for the second cutoff.

We compared differences in Encoding Phase-of-Firing Gain between narrow and broad-spiking neurons using the Kruskal–Wallis test, independently for each area. To clarify, we analyzed spike-LFP pairs here; thus, the same neuron may be included more than once.

**Assessing encoding linked to the temporal evolution of LFP.** We assessed how the sites we analyzed were related to the temporal evolution of the LFP in two ways, first, by assessing how the LFP power and phase changed with stimulus onset; and second, how encoding changed as a function of periods of particularly high or low beta power.

We determined how power and phase were distributed relative to the stimulus or reward onset. As for the spike-aligned analysis, we decomposed the LFP via the Fourier transform after Hanning tapering. We determined the spectral content 6–60 Hz frequency window, from [−2 2] s stepped every 5 ms.

Power was taken as the squared magnitude of the spectra representation. Power was normalized for 1/f noise. To determine the spectral peak across sites and epochs, we z-score normalized the power across all time points and epochs for each LFP individually. We report on the median of this normalized quantity.

To determine if there was evidence of phase resetting, we performed, for each LFP, a Rayleigh test at every point in time for every frequency, and extracted the Z statistic. We report on the median Raleigh Z value, with higher values related to a greater phase consistency across trials.

Finally, we assessed how encoding varied during burst periods[50,91]. We took an approach conceptually similar to Lundqvist and colleagues[91]. For each LFP channel, we first normalized the power for 1/f noise. Next, we averaged this signal in the same [10 25] Hz window we used for spike-aligned analysis above. Following this, we Z-scored beta power within each trial individually. Bursts were defined as periods where the normalized power exceeded 1.5 SD for a minimum of 3 cycles (=45 ms). The burst proportion was defined as the mean across trials at each point in time.

To assess how encoding varied as a function of burst periods, we separately selected spikes that either occurred within burst periods, or outside of burst periods, before calculating the EPFG as before. We only analyzed cell-LFP pairs with a minimum of 30 spikes after this selection.

## Data availability

Raw data is available upon reasonable request. A reporting summary for this Article is available as a Supplementary Information file. The source data underlying Figs. 4d–f and 5 are provided as a Source Data file. Source data are provided with this paper.

## Code availability

Code for analysis and reproduction of main conclusions is available online at https://github.com/att-circ-contrl/ana_phaseGain.

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

## Acknowledgements

This research was supported by a grant from the Canadian Institutes of Health Research CIHR Grant MOP_102482 (T.W.) and by the National Institute of Biomedical Imaging and Bioengineering of the National Institutes of Health under Award Number R01EB028161 (T.W.). The content is solely the responsibility of the authors and does not necessarily represent the official views of the Canadian Institutes of Health Research or the National Institutes of Health. The authors thank Drs. Jeffrey Schall, Andrew Tomarken, and Erin Calipari, for their insightful comments on drafts of the manuscript.

## Author contributions

Conceptualization, B.V. and T.W.; Methodology, B.V. and T.W.; Software, B.V. and T.W; Investigation, M.O. and T.W.; Writing – Original Draft, B.V. and T.W.; Writing – Review & Editing, all authors; Visualization, B.V. and T.W.; Supervision, T.W.

## Competing interests

The authors declare no competing interests.

## Additional information

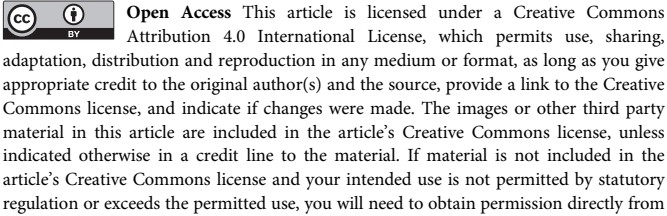

