## [Peer Review File · Nature Communications]

Reviewers' comments:

Reviewer #1 (Remarks to the Author):

In this study, the authors examine the involvement of neurons in 3 brain regions of interest – LPFC, ACC and Striatum – in the coding of 3 behavioral variables: choice outcome, outcome prediction errors, and outcome history. They also measured the synchrony of spiking with specific phases of LFP oscillations within and across areas (Pairwise-Phase Consistency), and tested whether the timing of such phase-synchronized spikes further contributes to the coding of 3 choice outcome variables. They report that a large proportion of neurons in the 3 areas encodes each of the outcome variables, and that the LFP phase-synchronization of spikes leads to an improvement in the coding of choice outcome variables. Overall, the study seems well designed and the questions motivating it are relevant. The main results could have potential impact, particularly those related to improvements in coding of choice variables due to LFP phase-synchronization. However, several of the analyses are too rough, and more thorough analyses will be required for the main claims to be convincing. Furthermore, in the current version of the manuscript, it is difficult to understand many details in the methods and results, since it lacks in accuracy, consistency, clarity and completeness with which results are reported and depicted in figures. These issues are described in detail below:

MAJOR COMMENTS:

1.) The most important claim of the authors is that the 3 outcome variables are better encoded by spikes when considering LFP-phase aligned spikes than when considering spikes independently of LFP phase. To show this, they built a quantitative method to estimate this gain in encoding, which they quantify as Encoding Phase Gain (EPG). However, it is difficult to get an idea of what values of EPG correspond in terms of more standard measures of information gain. It could be that this gain is statistically significant, but yet very small in magnitude, so small that it barely makes any difference to the brain. The authors should find a more standard way to measure, for each neuron, how much information there is about each outcome variable. Examples are: separability measured by ROC analysis, decoding (classification accuracy), or ANOVA PEV, etc. For the chosen metric, the authors should compute and compare the amount of information about each outcome variable in phase-aligned spikes vs. phase-randomized spikes. They can represent this gain as a percentage improvement in the amount of information. For example, the improvement can be represented as classification accuracy of 75% increased to 79%. Furthermore, they should show the values of all neurons (in addition to a just showing central tendency +/- dispersion measure [Fig. 4D]). The authors should then discuss how much this improvement matters to the brain based on their final measures of information increases.

2.) The authors talk about 3 functional "clusters" encoding Outcome, Prediction Error and Outcome History. These "clusters" are identified by how their weights of encoding of trial outcome by trial history behave. However, from the results reported, it is never obvious that there are 3 clearly segregated functional "clusters". Alternatively, it may be that the entire population of neurons homogeneously covers the entire space of combinations of strength of coding of Outcome, Prediction Error and Outcome History, but that neurons that are on the corners of that space preferentially encode one of the 3 variables. Can the authors measure how strongly each neuron encodes each of the 3 variables (i.e. the extent to which its activity profile matches each of the 3 types of coefficient distributions), and show whether neurons clearly fall in 1 of 3 separate clusters in a categorical manner, rather than just being homogeneously spread in that encoding space, with some neurons falling somewhere in the middle between 2 of the cluster encoding types. Reporting quantitative results of the cluster analysis and some graphical depiction of how the neurons cluster would help. However, if neurons don't clearly fall into separate clusters, the authors can modify their report to classify neurons into one of the 3 functional categories based on which of the 3 variables they preferentially (most strongly) encode.

3.) The manuscript is heavily based on trial outcome. However, the authors did not clearly describe all the possible errors that can be made by the monkeys and state which of those errors were included in their analysis and which were not (and their rationale for choosing those). For example, a monkey can make an error by responding to dimming of a distractor, or by making a saccade in the wrong direction (up/down), or by making a saccade when there isn't dimming of any stimulus. Furthermore, errors of target choice are different if the monkey makes them during a color reversal trial or in the middle of a block of same-target trials.

Stemming from the above: the manuscript does not sufficiently clarify how exactly the concept of "reward prediction error" is applied in the context of this study. To do this, the authors first need to clarify which errors were included in the analysis. Please add a few lines giving a precise definition of reward prediction error and clarifying what exactly is considered to be a reward prediction error in your task. As described above, an error of "target selection" are different if the monkey makes it during a color reversal trial or in the middle of a block of same-target trials. Reward prediction error would occur only during the former, not during the latter. For example, if the monkey responds to a distractor dimming during the middle of a block after several consecutive correct trials, it would not be considered to be a reward prediction error. Right? If so, then the authors can only use color reversal trials in this analysis, which probably means a low number of trials. Based on the experimental design, how many reversals did the authors have in one session? How did they ensure sufficient data to analyze given a potentially low number of trials? What was their solution?

4.) It seems important that the authors analyze synchrony as a function of time across the task trials. The authors should compare different measures of synchrony (PPC values, the percentage of neurons with significant synchrony, etc.) between different task periods for each area, particularly before vs. after the reward, and baseline vs. other task periods (especially reward). Does synchrony change between task periods, or would those measures be identical in an untrained macaque in resting state? Those two scenarios have very different mechanistic interpretations, and the authors should comment on that in their discussion.

5.) The manuscript is lacking a comparison of the strength of firing rate coding of Outcome, Prediction Error and Outcome History between areas. This is important because by the end of the manuscript, the authors compare the gain in encoding by phase between areas. The authors should report the strength of encoding of each of the above variables across all neurons in each of the 3 brain areas? A histogram showing the distribution of encoding of all neurons in each area would be best.

SPECIFIC COMMENTS:

6.) Line 139: "Neurons that encoded outcome variables in their firing rates showed similar phase synchrony as neurons not encoding reward outcome information". It would be useful to quantify this comparison. For example, you can compute the average 10-25 Hz PPC effect size for each pair of sites, and then statistically compare – and display – the distribution of values between coding sites and non-coding sites. From Fig. 3B, it appears that the effect size is higher in non-coding than coding sites. Whether it's higher or the same, the authors should comment on the interpretation of this in the discussion: This ubiquity of synchrony independent of function means that such synchrony is not specifically built to carry out particular functions, but rather ensures that the communication between areas is always phase-aligned, even during resting state.

7.) Results that are currently reported only across all pooled areas should be reported independently for different area pairs. For example: "Across all 141 pairs, 55% (4320/7938) showed significant phase synchronization within the 10-25 Hz range" (line 140). This percentage should be shown separately by pairs of areas, perhaps with a plot similar to 3D.

8.) Line 150: "Intra-areal LPFC and STR pairs showed statistically indistinguishable spike-phase

synchrony strength ($p=0.92$). This seems at odds with the results shown by Fig. 3D, where intra-areal LPFC synchrony strength is shown in orange (among the highest values) and intra-areal STR synchrony strength is shown in blue (among the lower values). Could the authors please explain?

9.) Figures 3B & S2: It is important to display the actual mean PPC values before displaying the effect size. Please add these figures.

10.) Line 152: "Similar findings were evident for the strength of inter-areal spike-LFP phase synchrony (Figure S2)."

What do the authors mean by similar findings? Similar is a comparative word. Similar between what and what? Intra-areal and inter-areal synchrony? Could the authors please follow up on this sentence to describe what exact findings you are referring to? Otherwise, readers will have no idea what exactly to look for in figure S2.

11.) Line 198: "we found that 139 (16%) spike-LFP pairs showed significant phase-of-firing encoding, i.e. these pairs encoded significantly more information in the phase of firing than in their firing rate alone"

The authors should split those 139 neurons by the brain areas where the spikes and LFPs came from. Are they preferentially from one particular area, or evenly distributed among all areas?

12. Line 205-207: When splitting EPG by area, does that mean that the authors considered the spikes from neurons in that area and classified their spikes based on phase from the other 2 areas? It is important to separate the results not only by the area where the spikes were, but also by the area with the phase-aligning LFP signal.

13.) Line 207: "EPG differences were more pronounced when selecting for each encoding metric the 25% of spike-LFP pairs with the largest EPG."

Please be more quantitative in this statement. What does "more pronounced" mean? Please report actual values or add a figure panel depicting this. The best way to do this is to show the distribution of EPG across all neurons, either showing all neurons as data points, or as a frequency histogram of EPG among all neurons.

14.) It makes it harder to compare Fig. 3E, F and G because they don't quite have the same y-axis scale. It is simple to make the axis the same between all 3 figures, and will allow the readers to compare all results without having to mentally rescale values.

15.) Line 836: "(A) We tested for functional connectivity between the ACC, LPFC, and STR."

Figure legends should describe the figures. The legend should state that the brain regions of interest are represented in the diagram as red areas in coronal brain sections of the cortex and striatum.

16.) Line 153: "For both monkeys neurons in ACC showed stronger spike synchronization compared to LPFC and STR spike output (Figure S3B)". According to Figure S3B, there is no significant difference between ACC and the other 2 areas in any of the monkeys other than ACC vs. LPFC in monkey HA. Please adjust the text to accurately represent the results shown in all figures.

17.) Line 204: "Similarly, EPG was evident for spike-LFP pairs with the spiking neuron in ACC ($p\sim 0$), in STR ($p=0.00028$), and in LPFC ($p=0.015$)".

Please always specify the statistical tests and the relevant statistics (t, F, etc.) whenever you report a p value. Please check this throughout the entire manuscript.

18.) Fig. S3C legend is incomplete. Please label the two axes in the polar plots and describe exactly what is being plotted. I'm guessing this is probably a frequency histogram of preferred phase across all neurons, correct? If so, please describe it as such.

19.) Please add a definition of the initialism "RPE" (line 103?). It seems to be an essential concept to the entire manuscript, but is never clearly defined.

20.) Typos:

Line 53: outcome-related should be hyphenated

Line 69: phase-synchronized

Line 72: neurons'

Reviewer #2 (Remarks to the Author):

Voloh et al report a very thorough study on the role of beta synchronization in carrying information regarding learning variables in inter-areal connections between anterior cingulate cortex, lateral prefrontal cortex and anterior striatum. This work addresses a timely question, and I found the results very interesting.

While overall a very compelling account, I am a little puzzled by how the non-preferred phase would carry the most information. Could the authors elaborate a bit more on potential mechanistic implications? The authors state that "Our results suggest that this updating can utilize spike-timing dependent plasticity mechanisms that are tuned to firing phases ~ 27 ms away from the preferred synchronization phase in the beta frequency band. How such a temporal organization in the beta band is used in the larger frontostriatal network will be an important question for future studies.", which, fair enough, but it would be good if they could at least provide some potential explanations or scenarios here.

To me it seems to go rather against the idea of the preferred phase providing ideal summation of spikes and therefore enhanced impact on the receiving site.

Regarding the nature of the beta rhythm, recent discussion of "bursting" nature of oscillatory dynamics suggests that beta may consist of transient events rather than ongoing, sustained oscillations. Averaging over trials with different latency bursts would give the suggestion of more sustained effects. It would be insightful if the authors can address the temporal pattern as well as trial-by-trial fluctuations of the underlying beta signal in that regard.

Did the individual regions show a beta peak in the power spectra? Where do these beta oscillations originate from? That is, are they generated by a particular node in the network (perhaps striatum?), is there a driving source here or local generation in each node?

It was not entirely clear to me whether all cells here fire at a beta rate, or whether there are also sites that fire at a different rate but nevertheless are coupled to a beta rhythm in the LFP.

Reviewer #3 (Remarks to the Author):

Voloh et al. present analyses of neural activity in frontal cortex (ACC & LPFC) and striatum in the outcome period of a primate reversal learning task in which subjects must track which of a pair of different colored stimuli is currently rewarded. They show that spikes synchronize with LFP beta oscillations across regions during outcome processing, and analyze how neural coding of learning related variables varies across beta oscillation phase. Their principal findings are that neuronal firing rates at different beta oscillation phases are differentially informative about learning variables, and that at least for some variables the phase of maximal encoding is different from the phase of peak firing, which they interpret as evidence for phase-based multiplexing. If shown convincingly these findings are important, but there are issues with the analyses that would need to be addressed before publication. The two key issues as I see it are the robustness of the statistics used to infer that encoding is modulated by beta phase, and the implementation of the underlying analysis used to infer what the neurons encode in the first place.

Phase encoding statistics:

Encoding strength was assessed using a regression model predicting spiking as a function of behavioral variables. This was applied separately to activity at different oscillation phases to assess encoding strength as a function of phase. A cosine was fit to the encoding strength across oscillation

phase to quantify the depth of modulation, which they term 'phase-of-firing gain' (PFG). To assess statistical significance, the authors estimated the expected PFG under the null hypothesis (which they term PFG_r) by randomly permuting phase labels on spikes and re-running the analysis many times. The PFG_r will not be zero because fitting a cosine to noise will always result in a positive amplitude, so for the stats to be robust it is important the PFG_r is estimated correctly. With any permutation based statistical method the choice of what is permuted is important because this determines the exact null hypothesis being tested. My concern with randomly permuting the phase labels on each spike independently is that it will remove any within trial correlation of spike phases and hence may be insufficiently conservative – effectively estimating the PFG distribution under the null hypothesis that there is no autocorrelation in spike activity, rather than the correct null hypothesis that there is no consistent phase relationship between behavioral variables and spike phase. As there is autocorrelation in spike activity (due to the oscillation if nothing else), the permutation test should preserve the autocorrelation of the spike activity within trial while randomizing phase relationships between behavioral variables and spike phase. As the behavioral variables only change from trial to trial, a simple permutation that achieves this is to permute the phase of all spikes on a given trial by the same random angle between 0 and 2π.

A second detail of these statistics is the width of the bins used to separate the spiking by phase. The authors sensibly choose these bin widths to equalize the number of spikes in each bin, but it is not specified what bin widths are used when running the permutation analysis to calculate PFG_r. Are the bin widths re-adjusted for each permutation to ensure equal numbers of spikes in each bin? Not doing so would presumably make the permutation test over-conservative.

A final question regarding these statistics is how the distribution of PFG under the null hypothesis (obtained via permutation) is used to assess significance. My understanding is that for most analyses the authors take the mean of this distribution (PFG_r) and subtract it from the PFG for the true data (PFG_e) to obtain what they term the Encoding Phase Gain (EPG), whose expectation is 0 under the null hypothesis. To assess statistical significance they compare the distribution of EPG across a population of neurons to zero using a Wilcoxon signed rank test. This is reasonable for asking questions of the population as a whole, but it would also be useful to assess how EPG is distributed across neurons (i.e. is there a sub-population of neurons with strong phase dependent encoding), and for what fraction of individual neurons the PFG is significantly different from that expected under the null distribution (P values for this can be calculated by comparing the PFG_e for a neuron with the distribution of PFG for that neuron across permutations).

The distribution of PFG across neurons is particularly important because on average the depth of modulation of encoding is weak – around 5 – 10% of the average encoding strength across the cycle. This raises questions about whether the modulation is functionally significant in addition to being statistically significant, as theoretical work suggests that efficient multiplexing with such phase modulated rate codes requires deep modulation (Akam & Kullman, PLOS Comp. Biol. 2012). Modulation depth assessed experimentally using spike-LFP phase coupling is likely to underestimate true modulation depths, so apparently weak modulation is not necessarily fatal for such claims, but this issue does need to be addressed head on in the discussion.

Basic characterisation of encoding:

The encoding regression analysis predicted neuronal firing as a function of recent outcomes (reward vs no reward) but does not take the choices made by the subject on these trials into account. This is a problem because the authors use the beta weights from this regression to infer whether neurons encode reward prediction errors (RPE), but clearly the influence of the previous trials reward on the current trials reward prediction error will depend on whether the previous trials choice is the same as the current trial choice. Indeed, the authors have previously done a much more in depth analysis of RPE coding in these brain regions on this task (Oemisch et al. Nat. Com. 2019), and showed that for the great major of RPE encoding neurons these RPEs were stimulus specific. More broadly, subjects

cannot solve this task using outcome information alone – they need to know which stimuli the outcomes were associated with, so it is rather unsatisfying to see an analysis of learning related activity which ignores half the information needed to solve the task.

Another question regarding the characterization of neurons is their clustering into three groups according to whether they represent current outcome, RPE or outcome history. This clustering plays a prominent role in subsequent analyses, so it is important that readers can clearly assess how distinct these putative clusters are. This is not clear because the clustering approach is rather unusual, making the metrics presented of cluster separation hard to interpret. An initial clustering was performed using a sensible distance metric (cosine distance on the regression beta weights). This clustering was repeated many times using bootstrap resampling of neurons, and a new similarity metric was created by assessing the number of times across the bootstrap samples each pair of neurons was clustered together. This new similarity metric was used for a final clustering. As far as I can tell it is this new similarity metric that is presented in figure S1B to show the similarity between and across clusters, but this metric is so far removed from the neurons actual encoding of task variables that it is not informative about cluster separation. The authors should show the similarity matrix (as figure S1B) using the underlying cosine distance metric, and also show the location of individual neurons colored coded by cluster assignment in the space defined by the beta weights for current and previous trial outcome (or a similar space defined by the actual beta weights) so the cluster separation can be visualized.

Other points:

- When analyzing the behavior using a regression analysis (Figure 1D) it would be more informative to show how previous choices and outcomes jointly predict current choice (as in e.g. Parker ... Witten, Nat. Neurosci. 2016), rather than how previous outcomes predict current outcomes.

- It was not completely clear to me how the encoding metrics defined in line 622 were used in evaluating EPG. I think that one metric was used for each neuron, with the metric chosen based on the neurons cluster assignment, but please clarify this.

- To give readers a more rounded picture of how beta oscillation activity interacts with the behavioral task, it would be useful to show how beta power varies across trial epoch and with correct/incorrect outcomes, and whether there is any phase resetting of the oscillation by trial events.

- Line 34 seems to suggest that striatum 'feeds back projections' to cortex – presumably this was referring to indirect connections, but the wording is unclear.

- Line 177 – for each neuron the frequency was selected that showed the greatest spike-LFP synchrony. How variable are these frequencies across neurons?

- The authors term the phase dependent encoding they identify as phase-of-firing coding, but it would be more accurately termed a phase modulated rate code, because neurons still encode behavioral variables by changes in their overall firing rate, but these changes are concentrated at particular phases.

- Figure 2-A: the raster plots do not add any value.

- Line 70: 'This phase-of-firing gain of encoding exceeded the firing rate code' This is misleading as the depth of modulation of rate coding across the cycle was much smaller than the average encoding strength .I think what the authors are trying to convey is that taking spike phase into account increased information available to the readout compared to just reading the firing rates. However, to really quantify this the authors would need to use a decoding approach – i.e. asking how much

information can be extracted by e.g. a linear classifier which either just has access to spike rates or has access to spike rates binned by phase. Such a decoding analysis would be a really nice complement to the encoding analysis and not a lot of work given that most of the preprocessing needed is already implemented.

Point-by-point replies (in blue font) to Reviewers' comments:

We highlight all changes made in red font of the revised text and reproduce the changed sections in our replies below.

Reviewer #1 (Remarks to the Author):

Comment: In this study, the authors examine the involvement of neurons in 3 brain regions of interest – LPFC, ACC and Striatum – in the coding of 3 behavioral variables: choice outcome, outcome prediction errors, and outcome history. They also measured the synchrony of spiking with specific phases of LFP oscillations within and across areas (Pairwise-Phase Consistency), and tested whether the timing of such phase-synchronized spikes further contributes to the coding of 3 choice outcome variables. They report that a large proportion of neurons in the 3 areas encodes each of the outcome variables, and that the LFP phase-synchronization of spikes leads to an improvement in the coding of choice outcome variables.

Overall, the study seems well designed and the questions motivating it are relevant. The main results could have potential impact, particularly those related to improvements in coding of choice variables due to LFP phase-synchronization. However, several of the analyses are too rough, and more thorough analyses will be required for the main claims to be convincing. Furthermore, in the current version of the manuscript, it is difficult to understand many details in the methods and results, since it lacks in accuracy, consistency, clarity and completeness with which results are reported and depicted in figures. These issues are described in detail below:

Reply: We thank the reviewer for acknowledging the potential impact of our study and for pointing us to various approaches to test, solidify and validate the main findings, and to being more comprehensive when reporting results for the three individual areas and the two monkeys.

Comment 1.) The most important claim of the authors is that the 3 outcome variables are better encoded by spikes when considering LFP-phase aligned spikes than when considering spikes independently of LFP phase. To show this, they built a quantitative method to estimate this gain in encoding, which they quantify as Encoding Phase Gain (EPG). However, it is difficult to get an idea of what values of EPG correspond in terms of more standard measures of information gain. It could be that this gain is statistically significant, but yet very small in magnitude, so small that it barely makes any difference to the brain. The authors should find a more standard way to measure, for each neuron, how much information there is about each outcome variable. Examples are: separability measured by ROC analysis, decoding (classification accuracy), or ANOVA PEV, etc. For the chosen metric, the authors should compute and compare the amount of information about each outcome variable in phase-aligned spikes vs. phase-randomized spikes. They can represent this gain as a percentage improvement in the amount of information. For example, the improvement can be represented as classification accuracy of 75% increased to 79%. Furthermore, they should show the values of all neurons (in addition to a just showing central tendency +/- dispersion measure [Fig. 4D]). The authors should then discuss how much this improvement matters to the brain based on their final measures of information increases.

Reply: We address these suggestions by adding several new analyses/results (with figures), enhancing the visualization of results, and adjusting the text to more explicitly convey the strength of the phase modulation effect.

We follow the reviewer's suggestion and added an ANOVA PEV based approach for GLMs, and also visualize the values for all neurons in adjusted and added figures. We specify the adjustments below.

First, we more explicitly describe that our main metric is the cosine amplitude of firing rate modulation, normalized by the mean encoding strength. It reflects the percent difference in firing rate differences for encoding peaks vs troughs. We further account for the upward bias in cosine amplitude estimation by recomputing this metric after randomly shuffling phases. Our key metric is thus a change in the phase/cosine modulation typically used for circular data, which is different to non-circular measures. The proposal of the reviewer to use standard measures does not apply in a straightforward way to circular modulation of rate (see below). We adjusted the text to clarify that our approach quantifies the ratio of cosine modulation for observed versus random shuffled rate modulation and how it can be interpreted, writing:

*“We estimated the strength of this phase modulation of rate encoding for each spike-LFP pair as the amplitude of a cosine that was fit to the phase-binned encoding metric, normalized by the mean encoding across phase bins, which we term the Phase-of-Firing Gain (PFG) (see **Methods**). We further accounted for the positive bias in cosine amplitude estimation by normalizing this quantity by the cosine amplitude obtained from fitting the phase-binned metric after randomly shuffling spike phases. We refer to this difference of the observed to the randomly shuffled phase modulation of encoding as the Encoding Phase-of-Firing Gain (EPFG; see **Methods**). This metric reflects an unbiased ratio of firing rate differences between preferred and anti-preferred encoding phase bins.”*

We increased the clarity of the procedure in the methods section and write:

“To determine if phase significantly added information above that of a phase-blind rate code, we opted for a randomization approach. For each cell, we first permuted the phase label of each spike, re-ran the GLM, re-fit a cosine and extracted the encoding phase-of-firing gain. This procedure was repeated 50 times, from which we obtain a distribution PFG_R of randomized encoding gains. For this procedure, because phase labels were permuted, the distribution of phases remains the same, and thus the bin widths need not be re-calculated. We report on the “excess” PFG, defined as the difference between empirical and the median of the randomized phase-of-firing gain, which we refer to in the manuscript at the Encoding Phase-of-Firing Gain (EPFG):

$$EPFG = PFG_E - \text{median}(PFG_R)$$

A positive value implies that encoding is modulated by phase above what would be expected by chance. To assess whether individual units showed significant encoding, we compared PFG_E against the null distribution PFG_R . Units were deemed significant at an alpha level of 0.05.”

Second, we followed the suggestion to also report a measure that has been used in other papers and opted to adopt the ANOVA PEV approach by quantifying the percent deviance explained (which is more appropriate for the Poisson GLM model we consider here) for each phase bin. The results with this method are similar, with the exception of the Outcome History cluster of cells. The revised methods describe the specific analysis steps by writing:

“The EPFG effectively quantifies the difference in mean firing rates between conditions, as a function of LFP phase. However, this does not necessarily imply that the information is easily decodable by other brain circuits. To address this question, we asked how much variance can be explained by the model fit to data in each phase bin. To this end, for each fit on each phase bin, we extracted the percent deviance explained (analogous to the ANOVA percent variance explained, but modified for a Poisson GLM). The percent deviance explained D^2 was calculated as (see ⁸²): $D^2 = 1 - \left(\frac{\text{Residual Deviance}}{\text{Null Deviance}} \right)$. The deviance for a Poisson distribution is defined as:

*Deviance = 2 * $\sum_i^n Y_i * \log\left(\frac{Y_i}{\lambda_i}\right) - (Y_i - \lambda_i)$. Where Y_i is the observed spike count on trial i , and λ_i is the predicted spike count. We then determined how D^2 varied as a function of phase using the same procedure as described above; namely, we fitted a cosine to the D^2 of each phase bin, extracted the amplitude, and compared it to a null distribution where phases have been permuted. We call this quantity the Encoding Phase-of-Firing Gain (D^2), or $EPFG_{D^2}$.*

The results with this method are summarized in a new supplementary Figure S7 and in a new section of the main text that includes various added statistical analyses with the heading “Robustness of phase-of-firing modulation of encoding”:

“The $EPFG$ is an effect size measure for how strong firing rate is modulated by LFP phase between conditions. However, it does not take into account the variability of firing rates across trials, leaving open the question of whether such mean firing rate changes may be effectively decoded. To address this question, we performed additional tests at the same beta frequencies at which neurons maximally synchronized. Firstly, we calculated how much the percent explained deviance varied as a function of phase, which quantifies how well the model fit the data with spikes extracted on individual phase bins. We term this quantity $EPFG_{D^2}$ (see Methods). We found that across areas and all spike-LFP pairs with significant encoding, $EPFG_{D^2}$ was significantly larger than chance (Wilcoxon signrank test, $p \sim 0$). $EPFG_{D^2}$ was significantly above chance for Outcome (Wilcoxon signrank test, $p \sim 0$) and RPE ($p = 0.001$) clusters, but not for spike-LFP pairs with neurons from the Outcome History cluster ($p = 0.24$) (Figure S7A).”

Third, we report the statistical results of two additional approaches. In one approach we preserve the within-trial firing phase distribution when constructing the randomization distribution (in response to reviewer 3), which provided similar results as our main shuffling statistics. In another approach we computed the $EPFG$ by normalizing to the overall neural firing rates, or encoding strength using all spikes. Both methods provided much higher $EPFG$ values, showing that our approach normalizes by the cosine modulation of the random phase-labeled shuffled rate is a more conservative approach (and might be considered to reflect a lower bound on the degree of modulation).

Fourth, in response to the reviewer we now show violin plots to show the more complete distribution of values of individual neurons beyond their means. We updated Figure 4,D,E,F,H, Figure S2E, S4B, S7A,B, and S9B.

Fifth, to convey more explicitly how much the phase modulation increases the encoding of variables beyond the firing rate encoding of the same variables, we performed two changes. First, as noted above, we have explicitly described how the $EPFG$ may be interpreted as an unbiased estimate of encoding beyond a firing rate code. Second, we determined the $EPFG$ by normalizing the PFG by two separate but relevant measures, the overall firing rate as well as the overall firing rate modulation. We summarize these findings by writing in the revised manuscript that:

“... , the $EPFG$ should be considered a lower bound on the degree of modulation. This is evident when normalizing the cosine modulation not by the null distribution of the cosine, but by the encoding strength determined using all spikes. With such a normalization, encoding strength is $\sim 0.61 \pm 0.03$, implying encoding is $\sim 61\%$ stronger on preferred vs anti-preferred phases. Similarly, normalizing the cosine modulation by the over-all firing rate of the cell, we obtained a median $EPFG$ of 0.18 ± 0.010 , implying that encoding is on average $\sim 18\%$ stronger on preferred rather than anti-preferred phases”

Comment 2.) The authors talk about 3 functional “clusters” encoding Outcome, Prediction Error and Outcome History. These “clusters” are identified by how their weights of encoding of trial outcome by trial history behave. However, from the results reported, it is never obvious that there are 3 clearly segregated functional “clusters”. Alternatively, it may be that the entire population of neurons homogeneously covers the entire space of combinations of strength of coding of Outcome, Prediction Error and Outcome History, but that neurons that are on the corners of that space preferentially encode one of the 3 variables. Can the authors measure how strongly each neuron encodes each of the 3 variables (i.e. the extent to which its activity profile matches each of the 3 types of coefficient distributions), and show whether neurons clearly fall in 1 of 3 separate clusters in a categorical manner, rather than just being homogeneously spread in that encoding space, with some neurons falling somewhere in the middle between 2 of the cluster encoding types. Reporting quantitative results of the cluster analysis and some graphical depiction of how the neurons cluster would help. However, if neurons don’t clearly fall into separate clusters, the authors can modify their report to classify neurons into one of the 3 functional categories based on which of the 3 variables they preferentially (most strongly) encode.

Reply: We thank the reviewer for pointing us to missing results and clarity of the clustering results. We address the comment in three steps.

First, we added to the text a clarification that the clustering analysis does not imply a categorical encoding of different variables. Rather, our analyses follow the logic that the reviewer pointed out last, namely, assigning to each neuron a “best” label (given what the rest of the population is doing). We now write:

*“The clustering does not preclude the possibility of a more continuous encoding space, but it statistically justifies focusing analysis on three sets of neurons with well distinguishable encoding pattern (see **Figure S2D**).*

Second, we added more explicit information on how we determined the degree to which clusters are discriminable by writing:

*“We used a clustering analysis to test whether the three types of outcome encoding were separable from each other and prevalent in each of the recorded brain areas (**Figure 2E-G, Figure S2A,B**). Clustering showed that neurons encoding each the three variables were statistically separable with reliable cluster assignments of neurons evident in an average Silhouette measure of cluster separability of 0.81 for LPFC, 0.57 for ACC, and 0.75 for STR (**Figure S2C**)⁴⁰.”*

High values imply that neurons tend to be clustered together (as expected from a “fragmented” space). Low values imply that the separation is less clear (consistent with a “continuous” space).

Third, we added figure panels in a new Fig. S2D to explicitly show how the coding of variables relate to the clustered neurons and refer to these results to show that the clusters group neurons with “well distinguishable encoding pattern (see **Figure S2D**)”.

We also added the detailed measures of the Silhouette for each area and cluster type in **Figure S2C**. Overall these results indicate that neurons were generally clustered appropriately (note the quadrant that each cluster falls into), but – as we write in the revised paper – “...does not preclude the possibility of a more continuous encoding space, ...”.

Comment 3.) The manuscript is heavily based on trial outcome. However, the authors did not clearly describe all the possible errors that can be made by the monkeys and state which of those errors were

included in their analysis and which were not (and their rationale for choosing those). For example, a monkey can make an error by responding to dimming of a distractor, or by making a saccade in the wrong direction (up/down), or by making a saccade when there isn't dimming of any stimulus. Furthermore, errors of target choice are different if the monkey makes them during a color reversal trial or in the middle of a block of same-target trials.

Stemming from the above: the manuscript does not sufficiently clarify how exactly the concept of "reward prediction error" is applied in the context of this study. To do this, the authors first need to clarify which errors were included in the analysis. Please add a few lines giving a precise definition of reward prediction error and clarifying what exactly is considered to be a reward prediction error in your task. As described above, an error of "target selection" are different if the monkey makes it during a color reversal trial or in the middle of a block of same-target trials. Reward prediction error would occur only during the former, not during the latter. For example, if the monkey responds to a distractor dimming during the middle of a block after several consecutive correct trials, it would not be considered to be a reward prediction error. Right? If so, then the authors can only use color reversal trials in this analysis, which probably means a low number of trials. Based on the experimental design, how many reversals did the authors have in one session? How did they ensure sufficient data to analyze given a potentially low number of trials? What was their solution?

Reply: We follow the reviewer with this point but do not agree with the main suggestion, which might be due to our under-specifications in the original manuscript of what errors and trials we used for analysis. We clarify this in the revised text.

Firstly, we need to apologize that we failed to mention in the original submission that we omitted from the analysis the first trial after a block reversal partly because that trial by definition was different to all other 2-trial sequences we analyzed (as it was an un-cued change in the reward rule and by definition erroneous). We added this information in the methods and write

"The trial immediately following a reversal event was not included in analysis."

Secondly, we clarify explicitly the type of errors we included in our analysis and write now:

"Correct responses were those that occurred according to the motion direction of the rewarded target in the correct response window, whereas errors were responses made to the incorrect, non-rewarded target, or in the incorrect response window in response to the distractor⁵."

We also clarify that we used only

"...outcomes from correct or erroneous choices, excluding fixation breaks..."

The reason to include only the errors that indicate a clear choice of the monkeys are twofold. The theoretically grounded reason is that these errors are those where it can be rightfully assumed that the animal wanted to make a correct choice, i.e. expected to be correct, but was not. In other words, these errors were based on an expectation – learned through a recent history of outcomes - that was only later violated by the lack of reward. This, by definition, is a reward prediction error. Another reason for including all 'choice errors' is that this maximizes the statistical power of our analysis. The animals are on average 80% correct given these errors, which leaves not many trials for fairly comparing errors and correct choices. We are currently working on experiments that will lead to more erroneous choices in order to gain statistical power for analyzing finer grained prediction error subtypes that is difficult with the 80%-correct reversal task we used in this study.

Thirdly, we highlight in the revised text that our definition of prediction errors is common in the error field. We added references to e.g. Asaad and Eskandars papers (e.g. Asaad, W. F. and E. N. Eskandar (2011). "Encoding of both positive and negative reward prediction errors by neurons of the primate lateral prefrontal cortex and caudate nucleus." *J Neurosci* 31(49): 17772-17787.) that also quantify prediction errors as we do. We added these references to the revised text to highlight that our definition of prediction error is not unusual and write:

"We found that during outcome processing, each area contained segregated ensembles of neurons whose firing rates encoded the current Outcome (firing differently for correct vs. errors), the Prediction Error of those outcomes (firing differently to an outcome when it differed versus was the same than in previous trial, as in e.g.^{31, 32}), and the recent Outcome History (increasing firing when the current outcome matched previous outcomes)."

Fourthly, we want to provide some direct empirical evidence to the reviewer that shows that reward prediction errors are high also during asymptotic performance (see Figure)

This figure quantifies shows the reward prediction errors (y-axis) for four example blocks of one of the monkey in our study for trials relative to the reversal. The reward prediction errors were computed with a (cross -validated) state-of-the-art attention-augmented reinforcement learning model. The figure and model are described in detail in Oemisch, M., et al. (2019). "Feature-specific prediction errors and surprise across macaque fronto-striatal circuits." *Nat Commun* 10(1): 176.

The example trials show that reward prediction errors are high for errors (open squares on top row in figure) that follow correct responses (filled squares), and they are high for correct trials that follow error trials. This illustration is meant to provide some additional rationale that our operational definition of prediction errors as trials with an outcome that differs to the previous trial is a valid operationalization. There is a longer history in the medial prefrontal cortex field that points to fact that all types of errors of commission are at their core unexpected outcomes (prediction errors) that trigger an adjustment in following trials to keep up or improve performance. Earlier evidence for this is described e.g. in Holroyd, C. B., et al. (2009). "When is an error not a prediction error? An electrophysiological investigation." *Cogn Affect Behav Neurosci* 9(1): 59-70.

Comment 4.) It seems important that the authors analyze synchrony as a function of time across the task trials. The authors should compare different measures of synchrony (PPC values, the percentage of neurons with significant synchrony, etc.) between different task periods for each area, particularly before vs. after the reward, and baseline vs. other task periods (especially reward). Does synchrony change

between task periods, or would those measures be identical in an untrained macaque in resting state? Those two scenarios have very different mechanistic interpretations, and the authors should comment on that in their discussion.

Reply: We follow the reviewer suggestion's, but we also want to highlight that our key findings are based on neurons when their firing rate contain outcome information. The firing rate and firing rate encoding of outcome/prediction error and outcome history increased sharply following the correct/error outcome signal (Supplementary Figure S2G and H), thus justifying our focus on this time period. The mechanistic sources of the neuronal synchronization appear therefore primarily important for interpreting and discussing the main findings. We made several adjustments to the revised text to address the raised points.

First, we report in the revised manuscript in a new Figure S3C the change in spike-LFP synchronization (PPC) between different task periods and report

*“[...] across all three areas, the strength of 15-25 Hz phase synchronization was statistically indistinguishable in the [-1 0] sec. pre-outcome period compared to the [0.1 1] sec. post-outcome period (Paired t-test, $abs(T) < 1.57$, $p > 0.12$; **Figure S3C**). The baseline period [-1 0] before stimulus onset) and the post-outcome period showed similar PPC values in ACC and STR ($abs(T) < 0.49$, $p > 0.6$), while LPFC showed stronger phase synchronization in the post-outcome period ($T = 2.82$, $p = 0.0049$; **Figure S3C**).”*

Second, we now compare in the revised manuscript the synchrony between areas and between neurons encoding different types of information We first report the strength and proportion of synchrony:

*“Across all ($n = 7938$) spike-LFP pairs, we found a pronounced peak of phase synchronization in the beta band (10-25 Hz) with neurons firing on average ~ 1.15 times more spikes on their preferred, average phase than at the opposite phase when considering the population average in the beta band, and 1.39 times more spikes on the preferred phase when selecting for each neuron the beta frequency with peak synchrony (**Figure 3B**, **Figure S3A**). Prominent beta-band synchrony was evident for neurons that encoded outcome variables in their firing rates and for those that did not show encoding (**Figure 3B**) with the peak synchrony being stronger for spike-LFP pairs at non-coding than coding sites (unpaired t-test, $T = 8.27$, $p \sim 0$; **Figure S3A**)*

*Overall 55% (4320/7938) of the spike-LFP pairs showed significant phase synchronization within the 10-25 Hz range (**Figure 3C**; Rayleigh test, $p < 0.05$, see Methods for prominence criteria), with similar proportions across all three areas (LPFC, 1506/2961, 50.9%; ACC, 1473/2442, 60.3%; STR, 1292/2524, 51.2%). Consistent with these results we found that the synchrony effect (the proportion of spikes at preferred over non-preferred phases) were similarly high for spike-LFP pairs with neurons encoding Outcome (1.37 ± 0.007), RPE (1.35 ± 0.013), and Outcome History (1.34 ± 0.011). There was only a trend for phase synchronization to be different between encoding clusters (ANOVA, $F = 2.8$, $p = 0.061$), which post-hoc analysis revealed to be driven primarily by differences between Outcome History and Outcome clusters ($p = 0.078$, multiple comparison corrected), rather than Outcome History and RPE ($p = 0.80$) or RPE and Outcome clusters ($p = 0.37$).*

[...]

*We found a trend for stronger between-area synchrony with spikes originating in STR, as compared to LPFC ($p = 0.058$) (**Figure 3D,F**). Testing for the reciprocity of beta-band phase synchrony showed that ACC spikes phases synchronized more strongly to LFP beta activity in the LPFC than vice versa ($p = 0.047$) (**Figure 3G**). LPFC and STR pairs showed statistically indistinguishable spike-phase synchrony strength ($p = 0.92$), as did ACC and STR pairs ($p = 0.26$).*

The findings were similar when inter-areal synchrony was analyzed separately at each frequency (Figure S3B)."

We also adjusted how we report the frequency specificity of the main phase of firing modulation and write now that we ...

"... tested whether EPFG was specific to the beta frequency band and how the strength of EPFG related to the strength of synchronization. First, we found that EPFG was strongest and significant at the population level in the same beta frequency band that showed the strongest spike-LFP synchronization (Figure 4G; Wilcoxon signrank test, $p < 0.05$)."

We added (also addressing a comment from reviewer 3) time resolved analysis for two main task epochs. First, we report LFP power around stimulus onset and around reward onset in a new Fig S8 and summarize these results by writing:

"... we found that band limited power in the beta band was a prominent and sustained component of the LFP after reward onset (Figure S8A), but without a reward-onset locked phase consistency (Figure S8B)."

We also clarify that our main effect does not scale with beta power and write:

"We found that overall the EPFG did not correlate with beta band power variations (Spearman rank correlation, $R = 0.050$, $p = 0.14$), but positively correlated with the overall firing rates of neurons (Spearman rank correlation, $R = 0.13$, $p \sim 0$)."

And we now report that our main phase-of-firing modulation was present during transient periods with high as opposed to low LFP oscillation amplitudes. We summarize this result by writing:

*"In addition to overall variations of power and firing rates, recent studies have shown that beta-band activity emerges in individual trials as transient bursts that can be linked to behavioral success in working memory and perceptual recognition paradigms^{48, 49, 50}. To test whether such burst occurrences may underlie the significant EPFG we report so far, we restricted the analysis of the EPFG to those beta band periods that were part of a suprathreshold, oscillatory burst event (see **Methods**). This analysis was performed for spike-LFP pairs when neurons fired sufficient (≥ 30) numbers of spikes per condition. The beta burst rate sharply increased after reward onset, as compared to a pre-reward onset period (see **Figure S9A**). We found that for spikes occurring within bursts, the median EPFG was 0.067 ± 0.034 , which was significantly above chance (**Figure 8B**; $n = 191$; Wilcoxon signrank test, $Z = 2.40$, $p = 0.016$). EPFG for spikes outside bursts was 0.038 ± 0.017 , which was also above chance (**Figure S9B**; $n = 769$; $Z = 4.51$, $p \sim 0$). Although encoding was higher inside rather than outside of bursts, this difference only approached a trend (Kruskal-Wallis test, $\chi^2 = 0.81$, $p = 0.057$)."*

We adjusted the discussion at many places to describe explicitly different mechanistic scenario's on how the phase of firing information can be realized in neuronal circuits (by local mechanisms or by inheriting spikes at specific spike times from other areas. We highlight all changes made to the text in red font. We conclude the main discussion by highlighting outstanding issues about the state specific feasibility of the phase of firing coding by writing:

"Future work needs to specify whether these scenarios are realized by beta rhythmically firing ensembles of neurons and how long-lasting and robust the encoding with phase-specific firing is

with regard to the overall firing rates and firing variability of individual neurons during active brain states.”

Comment 5.) The manuscript is lacking a comparison of the strength of firing rate coding of Outcome, Prediction Error and Outcome History between areas. This is important because by the end of the manuscript, the authors compare the gain in encoding by phase between areas. The authors should report the strength of encoding of each of the above variables across all neurons in each of the 3 brain areas? A histogram showing the distribution of encoding of all neurons in each area would be best.

Reply: We agree. In addition to the time resolved, Z-normalized encoding profiles for the different neuron types and brain areas in Supplementary Figure S2G, we also include the violin distribution plots in Figure S2E, and provide explicit statistical comparisons in the main text, where we now write:

*“ [...] the strength of encoding differed on the basis of area for Outcome cells (Kruskal Wallis test, $\chi^2=26.6$, $p=0$, with stronger encoding in ACC than LPFC or STR), as well as Outcome History encoding (χ^2 -test, $\chi^2=19.7$, $p=0$, with stronger encoding in ACC and LPFC than in STR), whereas the strength of RPE encoding was similar across areas ($\chi^2=2.49$, $p=0.29$) (see **Figure S2E** for all pair-wise comparisons).”*

Comment 6.) Line 139: “Neurons that encoded outcome variables in their firing rates showed similar phase synchrony as neurons not encoding reward outcome information”. It would be useful to quantify this comparison. For example, you can compute the average 10-25 Hz PPC effect size for each pair of sites, and then statistically compare – and display – the distribution of values between coding sites and non-coding sites. From Fig. 3B, it appears that the effect size is higher in non-coding than coding sites. Whether it’s higher or the same, the authors should comment on the interpretation of this in the discussion: This ubiquity of synchrony independent of function means that such synchrony is not specifically built to carry out particular functions, but rather ensures that the communication between areas is always phase-aligned, even during resting state.

Reply: We added the proposed test, visualize the result in Figure S3A, and provide the comparison of the strength in spike-LFP synchrony in the revised text. We summarize this by writing:

*“Prominent beta-band synchrony was evident for neurons that encoded outcome variables in their firing rates and for those that did not show encoding (**Figure 3B**) with the peak synchrony being stronger for spike-LFP pairs at non-coding than coding sites (unpaired t-test, $T=8.27$, $p=0$; **Figure S3A**).”*

We also added reporting that the magnitude of the phase of firing encoding gain does not scale with the strength of synchrony for RPE and Outcome History cells, but does for Outcome cells. We write:

“We found that the EPFG was uncorrelated with the PPC differences between conditions for neurons encoding RPE (Spearman correlation, $R=0.083$, $p=0.36$), or Outcome History ($R=0.074$, $p=0.41$). For Outcome encoding cells we found a moderate positive correlation with higher EPFG associated with larger differences in spike-LFP synchronization for correct versus error trials ($R=0.11$, $p=0.0067$).”

However, (as we had reported in our initial submission) we also found that the frequency at which synchronization happens matters for extracting the information. We added an analysis showing this:

*“Moreover, the frequency showing strongest spike-LFP synchronization and the frequency showing maximal encoding phase gain matched closely (median frequency ratio: 1 ± 0.01 SE; **Figure S11C**). This similarity of synchronization and encoding frequency did not differ on the basis of the functional designation (Kruskal-Wallis test, $\chi^2=0.047$, $p=0.98$), nor the area from which the spikes were sampled (Kruskal Wallis test, $\chi^2=0.53$, $p=0.77$).”*

We agree with the reviewer’s suggestion that the most ubiquitous spikes at the preferred phases are not the most informative – at least not for the learning variables that we studied. We followed the suggestion to expand on this point in the discussion. In one discussion section we now write e.g.

“[...] some prefrontal cortex neurons synchronize stronger at beta to posterior parietal areas when subjects choose one visual category over another⁴⁶, or when they maintain one object over another in working memory⁴⁷. These findings are broadly consistent with a communication-through-coherence schema where upstream senders are more coherent with downstream readers when they successfully compete for representation^{24, 64, 65}. Yet it has remained unclear how such a scheme may operate when multiple items must be multiplexed and transmitted in the same recurrent network^{7, 23, 28, 66, 67, 68}. Computationally, the multiplexing and the efficient transmission of information can operate in tandem when the temporal organization of activity is exploited at the sending and receiving site^{8, 26, 27, 69}. Consequently, selective synchronization between distal sites could be leveraged to enhance transmission selectivity, whereas temporally segregated information streams could enhance transmission capacity⁷⁰.”

We also added a new discussion paragraph that critically discusses the possible scenario’s that our findings regarding synchrony are consistent with. We now write:

“A caveat in interpreting the phase-modulated coding we report is that it is consistent with multiple coding schemes beyond a phase-based multiplexing⁷⁴. For example, spiking activity may be phase-synchronized in one condition but not another, or alternatively, conditions may be encoded on separate phases. Our results provide support for both coding schemes. Outcome cells resemble coding via an asynchronous code; that is to say, spike-LFP phase synchronization is higher in one condition than another, with no evidence of phase differences between conditions. On the other hand, RPE and Outcome History cells show evidence of phase-separation coding. These cells showed no significant difference in PPC between conditions, but did show a (near significant) trend towards firing on different phases.”

Comment 7.) Results that are currently reported only across all pooled areas should be reported independently for different area pairs. For example: “Across all 141 pairs, 55% (4320/7938) showed significant phase synchronization within the 10-25 Hz range” (line 140). This percentage should be shown separately by pairs of areas, perhaps with a plot similar to 3D.

Reply: We added the area specific results at several places. For the above example we now report:

*“Overall 55% (4320/7938) of the spike-LFP pairs showed significant phase synchronization within the 10-25 Hz range (**Figure 3C**; Rayleigh test, $p<0.05$, see **Methods** for prominence criteria), with similar proportions across all three areas (LPFC, 1506/2961, 50.9%; ACC, 1473/2442, 60.3%; STR, 1292/2524, 51.2%).”*

We added plots for area specific findings in the revised paper in all panels of Figure S2A-H, in new Supplementary figures including Figure S3B,C, in Figure S4 for each area and each monkey, in Figure S6 for the proportion of significant phase-of-firing encoding gain for all area pairs, in Figure S7 for showing

different statistical results, in Figure S8 for time resolved LFP power effects, in Figure S10 for encoding angles between all area pairs, and in Figure S11C for controlling for frequency consistency of encoding.

Comment 8.) Line 150: “Intra-areal LPFC and STR pairs showed statistically indistinguishable spike-phase synchrony strength ($p=0.92$)”. This seems at odds with the results shown by Fig. 3D, where intra-areal LPFC synchrony strength is shown in orange (among the highest values) and intra-areal STR synchrony strength is shown in blue (among the lower values). Could the authors please explain?

Reply: Apologies, this was a mistake. The word “Intra-areal” was erroneous there - we are comparing the off-diagonal elements from Figure 3D, specifically the top right and bottom left elements. We have removed this word (and a grateful this was corrected).

Comment 9.) Figures 3B & S2: It is important to display the actual mean PPC values before displaying the effect size. Please add these figures.

Reply: We are not sure what additional plots with PPC values would add. The effect size is a linear scaling of the PPC values. We describe the equation explicitly in the methods section by writing:

$$\text{Effect size} = \frac{1+2*\text{sqrt}(PPC)}{1-2*\text{sqrt}(PPC)},$$

“The PPC effect size was determined as previously reported^{12, 13}.”

We show raw PPC values for all examples (Fig S5). We so far thought that reporting the effect size could be more informative than PPC values and would hope to see this more in other papers too (that still typically do not report the effect size measures). For space considerations we refrain from adding the plots.

Comment 10.) Line 152: “Similar findings were evident for the strength of inter-areal spike-LFP phase synchrony (Figure S2).” What do the authors mean by similar findings? Similar is a comparative word. Similar between what and what? Intra-areal and inter-areal synchrony? Could the authors please follow up on this sentence to describe what exact findings you are referring to? Otherwise, readers will have no idea what exactly to look for in figure S2.

Reply: Thank you for pointing us to this unclear sentence. We adjusted the paragraph and now write with more clarity and specific results:

“Consistent with these results we found that the synchrony effect (the proportion of spikes at preferred over non-preferred phases) were similarly high for spike-LFP pairs with neurons encoding Outcome (1.37 ± 0.007), RPE (1.35 ± 0.013), and Outcome History (1.34 ± 0.011). There was only a trend for phase synchronization to be different between encoding clusters (ANOVA, $F=2.8$, $p=0.061$), which post-hoc analysis revealed to be driven primarily by differences between Outcome History and Outcome clusters ($p=0.078$, multiple comparison corrected), rather than Outcome History and RPE ($p=0.80$) or RPE and Outcome clusters ($p=0.37$).”

Comment 11.) Line 198: “we found that 139 (16%) spike-LFP pairs showed significant phase-of-firing encoding, i.e. these pairs encoded significantly more information in the phase of firing than in their firing rate alone”

The authors should split those 139 neurons by the brain areas where the spikes and LFPs came from. Are they preferentially from one particular area, or evenly distributed among all areas?

Reply: To address this comment we provide in the revised manuscript the area specific results, adding Figure S6 to show all possible area combinations and summarizing them in the main text by writing:

*“...Similarly, spike-LFP pairs with spikes from an ACC neuron were more likely to show individually significant EPFG (χ^2 test, $\chi^2=17.7$, $p=0.0014$; **Figure S6**).”*

Comment 12. Line 205-207: When splitting EPG by area, does that mean that the authors considered the spikes from neurons in that area and classified their spikes based on phase from the other 2 areas? It is important to separate the results not only by the area where the spikes were, but also by the area with the phase-aligning LFP signal.

Reply: We agree and added and include a new analysis conditioning our results on the LFP-site, finding that:

*“When considering encoding strength on the basis of the LFP site of the spike-LFP pairs EPFG was above chance in each of the three areas (**Figure 4F**; ACC, $Z=5.02$, $p\sim 0$; LPFC, $Z=5.62$, $p\sim 0$; and STR, $Z=5.8$, $p\sim 0$), but did not vary by the LFP area (Kruskal Wallis test, $\chi^2=0.192$, $p=0.91$).”*

Comment 13.) Line 207: “EPG differences were more pronounced when selecting for each encoding metric the 25% of spike-LFP pairs with the largest EPG.” Please be more quantitative in this statement. What does “more pronounced” mean? Please report actual values or add a figure panel depicting this. The best way to do this is to show the distribution of EPG across all neurons, either showing all neurons as data points, or as a frequency histogram of EPG among all neurons.

Reply: Yes, sorry for these missing numbers. We added statistical results and show the complete distribution in the form of violin plots and by adding figures for each monkey. We write:

*“This selection revealed stronger EPFG encoding of RPE compared to Outcome ($\chi^2=11.3$, $p\sim 0$) and Outcome History ($\chi^2=11.3$, $p\sim 0$). It also provided additional confirmation that EPFG was larger for neurons in ACC than in LPFC ($\chi^2=10.4$, $p=0.0013$), with a similar trend for STR ($\chi^2=2.41$, $p=0.12$). Likewise, EPFG did not vary on the basis of LFP area (Kruskal Wallis, $\chi^2=0.192$, $p=0.91$). These results were similar in each monkey (**Figure S4B**).”*

Comment 14.) It makes it harder to compare Fig. 3E, F and G because they don't quite have the same y-axis scale. It is simple to make the axis the same between all 3 figures, and will allow the readers to compare all results without having to mentally rescale values.

Reply: We have adjusted the figure axes to aid in comparison between panels.

Comment 15.) Line 836: “(A) We tested for functional connectivity between the ACC, LPFC, and STR.” Figure legends should describe the figures. The legend should state that the brain regions of interest are represented in the diagram as red areas in coronal brain sections of the cortex and striatum.

Reply: We have adjusted the figure legend accordingly.

Comment 16.) Line 153: “For both monkeys neurons in ACC showed stronger spike synchronization compared to LPFC and STR spike output (Figure S3B)”. According to Figure S3B, there is no significant difference between ACC and the other 2 areas in any of the monkeys other than ACC vs. LPFC in monkey HA. Please adjust the text to accurately represent the results shown in all figures.

Reply: We double checked the figure panels and statistics and updated the text to accurately reflect what is shown in the figure (which was S3 and now is S4): ACC showed the strongest synchrony on average in both animals, both within and between areas. We write:

*“For both monkeys, neurons in ACC showed the strongest spike synchronization compared to neurons from LPFC and STR (the area difference is significant in monkey HA and trends the same way in monkey KE; see **Figure S4A**).”*

Comment 17.) Line 204: “Similarly, EPG was evident for spike-LFP pairs with the spiking neuron in ACC ($p \sim 0$), in STR ($p=0.00028$), and in LPFC ($p=0.015$)”.

Please always specify the statistical tests and the relevant statistics (t, F, etc.) whenever you report a p value. Please check this throughout the entire manuscript.

Reply: Thank you for pointing out this missing information. We have carefully reviewed the manuscript and added relevant statistics where applicable.

Comment 18.) Fig. S3C legend is incomplete. Please label the two axes in the polar plots and describe exactly what is being plotted. I’m guessing this is probably a frequency histogram of preferred phase across all neurons, correct? If so, please describe it as such.

Reply: Yes apologies for this. We updated the figure legends (which was S3C and now is S4C). It was the polar plot histogram:

“(C) Polar histograms of the preferred firing phase (upper panels) and maximal encoding phase (bottom panels) for each encoding cluster.”

Comment 19.) Please add a definition of the initialism “RPE” (line 103?). It seems to be an essential concept to the entire manuscript, but is never clearly defined.

Reply: Thank you for catching this oversight. We have defined RPE when we first describe it.

Comment 20.) Typos:

Line 53: outcome-related should be hyphenated - **corrected**

Line 69: phase-synchronized - **corrected**

Line 72: neurons’ - **corrected**

Thank you for pointing us to these.

Reviewer #2 (Remarks to the Author):

Comment 1. Voloh et al report a very thorough study on the role of beta synchronization in carrying information regarding learning variables in inter-areal connections between anterior cingulate cortex, lateral prefrontal cortex and anterior striatum. This work addresses a timely question, and I found the results very interesting.

Reply: We very much thank the reviewer for the positive reception.

Comment 2. While overall a very compelling account, I am a little puzzled by how the non-preferred phase would carry the most information. Could the authors elaborate a bit more on potential mechanistic implications? The authors state that “Our results suggest that this updating can utilize spike-timing dependent plasticity mechanisms that are tuned to firing phases ~27 ms away from the preferred synchronization phase in the beta frequency band. How such a temporal organization in the beta band is used in the larger frontostriatal network will be an important question for future studies.”, which, fair enough, but it would be good if they could at least provide some potential explanations or scenarios here. To me it seems to go rather against the idea of the preferred phase providing ideal summation of spikes and therefore enhanced impact on the receiving site.

Reply: We agree and are grateful for the suggestion to extend the discussion about the mechanistic aspects. We added in the discussion an explicit consideration of two possible scenarios that might underlie the phase offset between encoding and preferred phases. They are both linked to the recent models of the Kopell group showing phase specific multiplexing of different input streams into a circuit; we write:

*“The parallel coding of information at a common beta rhythm in these models provides a qualitative proof of concept about phase specific encoding of multiple types of inputs in larger beta rhythmic ensembles, and suggests a possible mechanistic realization of enhanced encoding by the phase of firing in the beta band²³. Moreover, these models^{23, 41} also suggest possible reasons why encoding phases and the average, preferred spiking phases can differ. In our study RPE encoding was maximal for spikes that occurred 27 ms away from the preferred beta phase at which most spikes of the neurons were elicited. In the context of these models, a phase offset could indicate that RPE’s are part of an input stream that is arriving already with a delay to the major beta rhythmic input stream that these neurons sees. For example, input carrying prediction error information might arrive from the ventral tegmental area while the dominant beta rhythmic firing (that determines the mean phase) might be based on local intracortical mechanisms coupled to other cortical areas. Consistent with such a scenario, a prior rodent study⁷⁴ has shown that the phase of phase synchronous prefrontal cortex neurons shifted with the learning of new reward locations, possibly with a dopaminergic influence from the ventral tegmental area on the phase of spike-LFP synchrony. Alternatively, a 27 ms phase offset for encoding prediction error information might have a local origin with the delay reflecting the computation of the error in prediction based on input that carry the prediction itself. This scenario gains plausibility when considering that a prediction error reflects a transformation of two signals, i.e. it is the difference of the expected value and the received outcome. This transformation will take time. In the temporal domain, this delay is likely reflected in a latency difference with prediction error signals emerging typically after outcomes are processed (which we found, **Figure S2G, H**). In a*

recurrent circuit, this delay in computing an error might be additionally be reflected in a phase offset. According to this view the 27 ms offset in maximal encoding of RPE indicates a local transformation of two input streams (predicted value and outcome) into their difference (the error in value prediction). Future work needs to specify whether these scenarios are realized by beta rhythmically firing ensembles of neurons and how long-lasting and robust the encoding with phase-specific firing is with regard to the overall firing rates and firing variability of individual neurons during active brain states.”

Comment 3. Regarding the nature of the beta rhythm, recent discussion of “bursting” nature of oscillatory dynamics suggests that beta may consist of transient events rather than ongoing, sustained oscillations. Averaging over trials with different latency bursts would give the suggestion of more sustained effects. It would be insightful if the authors can address the temporal pattern as well as-trial-by-trial fluctuations of the underlying beta signal in that regard.

Reply: We agree and added the suggested analysis to the revised manuscript with two new figures (S8 and S9).

First, we show the average temporal evolution of trial averaged beta activity in ACC, LPFC and STR in a new supplementary Figure S8 and summarize that ...

“...we found that band limited power in the beta band was a prominent and sustained component of the LFP after reward onset (Figure S8A) but without a reward-onset locked phase consistency (Figure S8B).”

Second, we quantify the transient nature of beta activity by reporting the rate of beta bursts across trials similar to previous studies. We found that the burst rate increases during the reward epoch and that the phase-of-firing encoding gain is evident for spikes inside and outside of these bursts. We summarize these results by writing:

“In addition to overall variations of power and firing rates, recent studies have shown that beta-band activity emerges in individual trials as transient bursts that can be linked to behavioral success in working memory and perceptual recognition paradigms^{46, 47, 48}. To test whether such burst occurrences may underlie the significant EPFG we report so far, we restricted the analysis of the EPFG to those beta band period that were part of a suprathreshold, oscillatory burst event (see Methods). This analysis was performed for spike-LFP pairs when neurons fired sufficient numbers of spikes (≥ 30) per condition. The beta burst rate sharply increased after reward onset, as compared to a pre-reward onset period (see Figure S9A). We found that for spikes occurring within bursts, the median EPFG was 0.067 ± 0.034 , which was significantly above chance (Figure 8B; $n=191$; Wilcoxon signrank test, $Z=2.40$, $p=0.016$). EPFG for spikes outside bursts was 0.038 ± 0.017 , which was also above chance (Figure S9B; $n=769$; $Z=4.51$, $p\sim 0$). Although encoding was higher inside rather than outside of bursts, this difference only approached a trend (Kruskal-Wallis test, $\chi^2 = 20.81$, $p=0.057$).”

Comment 4. Did the individual regions show a beta peak in the power spectra? Where do these beta oscillations originate from? That is, are they generated by a particular node in the network (perhaps striatum?), is there a driving source here or local generation in each node?

Reply: Yes, each of the regions shows the beta activity. We added this more explicitly at several places in the revised manuscript.

In the new supplementary Figure S8 we show that these areas have a similar temporal evolution and frequency of beta LFP activity by writing (see also previous comment) that:

“...we found that band limited power in the beta band was a prominent and sustained component of the LFP after reward onset (Figure S8A) but without a reward-onset locked phase consistency (Figure S8B).”

In a new supplementary Figure S3 we show the spike-LFP spectra for each area, revealing the same low beta frequency specific peak in synchrony and summarize this result as:

*“Across all ($n=7938$) spike-LFP pairs, we found a pronounced peak of phase synchronization in the beta band (10-25 Hz), with neurons firing on average ~ 1.15 times more spikes on their preferred, average phase than at the opposite phase when considering the population average in the beta band (**Figure 3B**), and ~ 1.39 times more spikes on the preferred phase when selecting for each neuron the beta frequency with peak synchrony (**Figure S3A**). Prominent beta-band synchrony was evident for neurons that encoded outcome variables in their firing rates and for those that did not show encoding (**Figure 3B**), with the peak synchrony being stronger for cell-LFP pairs with non-coding rather than coding cells (unpaired t -test, $T=8.27$, $p\sim 0$; **Figure S3A**). Overall 55% (4320/7938) of the spike-LFP pairs showed significant phase synchronization within the 10-25 Hz range (**Figure 3C**; Rayleigh test, $p<0.05$, see **Methods** for prominence criteria), with similar proportions across all three areas (LPFC, 1506/2961, 50.9%; ACC, 1473/2442, 60.3%; STR, 1292/2524, 51.2%).”*

To be sure that it is the same frequency that carries the phase of firing encoding gain, we note our previous analysis of the frequency consistency of the main effects, writing:

*“... [that we] tested whether the dissociation of spike- and encoding- phases is not based on possible systematic phase shifts due to differences in the peak oscillation frequencies within the beta band. We validated that this was not the case and found that the three sets of neuronal encoding clusters synchronized on average at the same ~ 15 Hz center frequency (Kruskal Wallis, $\chi^2=0.95$, $p=0.62$; **Figure S11A**), and that they showed maximal phase-of-firing encoding at similar frequencies (also ~ 15 Hz) (Kruskal Wallis test, $\chi^2=0.39$, $p=0.82$; **Figure S11B**). Moreover, the frequency showing strongest spike-LFP synchronization and the frequency showing maximal encoding phase gain matched closely (median frequency ratio: 1 ± 0.01 SE; **Figure S11C**). This similarity of synchronization and encoding frequency did not differ on the basis of the functional designation (Kruskal-Wallis test, $\chi^2=0.047$, $p=0.98$), nor by the area from which the spikes were sampled (Kruskal Wallis test, $\chi^2=0.53$, $p=0.77$).”*

We cannot be sure about the origin of the beta oscillation in our recordings. We plan on addressing this issue with laminar specific cortical recordings, and cell type specific analysis of beta activity in a separate set of studies.

The circuit models we refer to in the discussion realize beta through cell-specific properties in deep cortical layers, but in the striatum beta activity will be driven by a different set of properties of medium spiny neurons (that is even slightly different for D1 and D2 medium spiny neurons). We feel that a fair discussion of the possible local and long-range mechanisms that could sustain beta would at this stage add speculation to the already long (~ 2250 word) discussion that is rather remote from the main mechanistic and functional questions that we aim to discuss to make sense of the main results.

Comment 5. It was not entirely clear to me whether all cells here fire at a beta rate, or whether there are also sites that fire at a different rate but nevertheless are coupled to a beta rhythm in the LFP.

Reply: We added a more explicit section to the results with a new supplementary Figure S3B to show that beta synchrony was the prevailing and most prominent frequency specific firing effect for most (but not all) cells. We write in the revised paper (see also comment 4):

*“Across all ($n=7938$) spike-LFP pairs, we found a pronounced peak of phase synchronization in the beta band (10-25 Hz), with neurons firing on average ~ 1.15 times more spikes on their preferred, average phase than at the opposite phase when considering the population average in the beta band (**Figure 3B**), and ~ 1.39 times more spikes on the preferred phase when selecting for each neuron the beta frequency with peak synchrony (**Figure S3A**). Prominent beta-band synchrony was evident for neurons that encoded outcome variables in their firing rates and for those that did not show encoding (**Figure 3B**), with the peak synchrony being stronger for cell-LFP pairs with non-coding rather than coding cells (unpaired t -test, $T=8.27$, $p\sim 0$; **Figure S3A**). Overall 55% (4320/7938) of the spike-LFP pairs showed significant phase synchronization within the 10-25 Hz range (**Figure 3C**; Rayleigh test, $p<0.05$, see **Methods** for prominence criteria), with similar proportions across all three areas (LPFC, 1506/2961, 50.9%; ACC, 1473/2442, 60.3%; STR, 1292/2524, 51.2%).*

We also found that the main phase-of-firing encoding effect was limited to the beta band (we did not find a separate lower frequency peak of encoding gain). We show the frequency specificity in Figure 4G and write:

“...we found that EPFG was strongest and significant at the population level in the same beta frequency band that showed the strongest spike-LFP synchronization (Figure 4G; Wilcoxon signrank test, $p<0.05$).”

Reviewer #3

(Remarks to the Author):

Voloh et al. present analyses of neural activity in frontal cortex (ACC & LPFC) and striatum in the outcome period of a primate reversal learning task in which subjects must track which of a pair of different colored stimuli is currently rewarded. They show that spikes synchronize with LFP beta oscillations across regions during outcome processing, and analyze how neural coding of learning related variables varies across beta oscillation phase. Their principal findings are that neuronal firing rates at

different beta oscillation phases are differentially informative about learning variables, and that at least for some variables the phase of maximal encoding is different from the phase of peak firing, which they interpret as evidence for phase-based multiplexing. If shown convincingly these findings are important, but there are issues with the analyses that would need to be addressed before publication. The two key issues as I see it are the robustness of the statistics used to infer that encoding is modulated by beta phase, and the implementation of the underlying analysis used to infer what the neurons encode in the first place.

Reply: We thank the reviewer for acknowledging the importance of this topic. We outline below how we addressed the concern of robustness and analyses implementation.

Comment 1. Phase encoding statistics:

Encoding strength was assessed using a regression model predicting spiking as a function of behavioral variables. This was applied separately to activity at different oscillation phases to assess encoding strength as a function of phase. A cosine was fit to the encoding strength across oscillation phase to quantify the depth of modulation, which they term ‘phase-of-firing gain’ (PFG). To assess statistical significance, the authors estimated the expected PFG under the null hypothesis (which they term PFG_r) by randomly permuting phase labels on spikes and re-running the analysis many times. The PFG_r will not be zero because fitting a cosine to noise will always result in a positive amplitude, so for the stats to be robust it is important the PFG_r is estimated correctly. With any permutation based statistical method the choice of what is permuted is important because this determines the exact null hypothesis being tested. My concern with randomly permuting the phase labels on each spike independently is that it will remove any within trial correlation of spike phases and hence may be insufficiently conservative – effectively estimating the PFG distribution under the null hypothesis that there is no autocorrelation in spike activity, rather than the correct null hypothesis that there is no consistent phase relationship between behavioral variables and spike phase. As there is autocorrelation in spike activity (due to the oscillation if nothing else), the permutation test should preserve the autocorrelation of the spike activity within trial while randomizing phase relationships between behavioral variables and spike phase. As the behavioral variables only change from trial to trial, a simple permutation that achieves this is to permute the phase of all spikes on a given trial by the same random angle between 0 and 2π.

Reply: We fully agree and performed the proposed more conservative randomization approach that preserved the autocorrelation. The results of this analysis are visualized in a new supplementary Figure S7B. We describe the rationale (thanks to the reviewer for pointing us to this) and results of the analysis by writing in the revised manuscript:

“In a second approach, we tested whether the EPFG [Encoding Phase-of-Firing Gain] is evident even when the statistical testing preserves the within-trial correlation of spike phases. So far, we tested for significance of EPFG by constructing a random distribution that shuffled all spike phases irrespective of the trials in which they occurred. While this preserves the overall degree of synchrony, it destroys any within-trial correlation of spikes. When constructing null distributions by randomly perturbing the phase of spikes on each trial, we found an overall significant EPFG of 0.080 ± 0.018 (Wilcoxon signrank test, $Z=7.6$, $p \sim 0$). As in the other statistics, EPFG was significant for Outcome ($p \sim 0$), Outcome History ($p=0.002$), and RPE (0.039) (Figure S7B). Similarly, phase-of-firing modulation significantly differed by area (Kruskal Wallis test, $p=0.032$), with spike-LFP pairs with spikes of neurons in ACC showing higher EPFG than LPFC ($p=0.012$) and a trend for higher EPFG in ACC than STR ($p=0.069$) (Figure S7B). Thus, the observed phase gain for the firing rate information is evident even when within-trial autocorrelation is preserved.”

We added the proposed results as a second statistical test to show the robustness of our findings to the way the randomization distribution is constructed. In our original choice of randomization, we followed the goal to preserve overall spike-LFP phase synchrony. In other words, in our approach, the degree of spike-phase synchrony is preserved, but (as the reviewer pointed out) any within-trial relationship to the behavioral variables is destroyed.

Comment 2. A second detail of these statistics is the width of the bins used to separate the spiking by phase. The authors sensibly choose these bin widths to equalize the number of spikes in each bin, but it is not specified what bin widths are used when running the permutation analysis to calculate PFG_r. Are the bin widths re-adjusted for each permutation to ensure equal numbers of spikes in each bin? Not doing so would presumably make the permutation test over-conservative.

Reply: For our main analysis, since we simply permuted the phase labels, the distribution of phases remained the same, and thus we did not re-determine phase bins. We note this in the methods, writing:

“Note that since phase labels were permuted, the distribution of phases remains the same, and thus the bin-widths need not be re-calculated.”

For the new randomization approach the reviewer suggested (involving randomizing phases across trials), since the phase distribution changes, we re-computed phase bins for each randomization. We describe this explicitly in the methods:

“... The procedure described above destroys any within-trial correlation between spike phases. Thus, in a related analysis, we determined PFG_R by adding a random phase in the range [0 2π] to all spikes within a single trial, thus preserving their correlation structure. In this case, the phase bin widths were re-calculated for every permutation.”

Finally, for the sake of completeness, we also report that the number of bins used did not affect the main findings by writing:

“..., we considered the stability of encoding. Encoding designation was stable across phase bins, with ~90% of spike-LFP pairs exhibiting similar beta coefficient signs across all phase bins, and was not dependent on the number of phase bins used (no correlation of EPFG with [4, 6, 8, and 10] number of bins (Spearman rank correlation, R=0.023, p=0.18).”

Comment 3. A final question regarding these statistics is how the distribution of PFG under the null hypothesis (obtained via permutation) is used to assess significance. My understanding is that for most analyses the authors take the mean of this distribution (PFG_r) and subtract it from the PFG for the true data (PFG_e) to obtain what they term the Encoding Phase Gain (EPG), whose expectation is 0 under the null hypothesis. To assess statistical significance they compare the distribution of EPG across a population of neurons to zero using a Wilcoxon signed rank test. This is reasonable for asking questions of the population as a whole, but it would also be useful to assess how EPG is distributed across neurons (i.e. is there a sub-population of neurons with strong phase dependent encoding), and for what fraction of individual neurons the PFG is significantly different from that expected under the null distribution (P values for this can be calculated by comparing the PFG_e for a neuron with the distribution of PFG for that neuron across permutations).

Reply: We follow the reviewer and agree. First, we added more details about the individual cell results in the revised manuscript where we write:

“Of the 877 spike-LFP pairs that significantly synchronized in the 10-25 Hz band and encoded information in their firing rate, we found that 139 (16%) spike-LFP pairs showed significant EPFG, i.e. these pairs encoded significantly more information when taking into account the phase of firing than their average, phase-blind firing rate (randomization test, $p < 0.05$).”

Second, we added in response to the reviewer suggestion (as well as for reviewer 1) the complete adjacency matrix of data showing the proportion of significant spike-LFP pairs per area combination in a new supplementary Figure S6. We refer to it by writing in the revised text:

*“..., spike-LFP pairs with spikes from an ACC neuron were more likely to show individually significant EPFG (χ^2 test, $\chi^2=17.7$, $p=0.0014$; **Figure S6**).”*

We added the statistical details in the methods section and now write:

“To assess whether individual units showed significant encoding, we compared PFG_E against the null distribution of PFG_R . Units were deemed significant at an alpha level of 0.05.”

Comment 4. The distribution of PFG across neurons is particularly important because on average the depth of modulation of encoding is weak – around 5 – 10% of the average encoding strength across the cycle. This raises questions about whether the modulation is functionally significant in addition to being statistically significant, as theoretical work suggests that efficient multiplexing with such phase modulated rate codes requires deep modulation (Akam & Kullman, PLOS Comp. Biol. 2012). Modulation depth assessed experimentally using spike-LFP phase coupling is likely to underestimate true modulation depths, so apparently weak modulation is not necessarily fatal for such claims, but this issue does need to be addressed head on in the discussion.

Reply: We agree with this caveat that statistical significance does not allow inferring functional relevance. To discuss this point explicitly we have made various adjustments to the text and hope these adjustments are used by readers to judge fairly about the possible implications of our findings.

First, to directly address the need to describe the effect size of the phase modulation and to point out that our measure is likely reflecting a lower bound we added an analysis and report:

“The EPFG is a normalized quantity that accounts for the fact that simply fitting a cosine will result in positive amplitudes, implying that a cosine amplitude on its own has an upwards bias. A similar bias is evident in the null distribution of cosine amplitudes. As a consequence, the EPFG should be considered a lower bound on the degree of modulation. This can be seen when normalizing the cosine amplitude modulation not by the null distribution cosine amplitude but instead by the firing rate of the cell. With such a normalization by the firing rate of the cell we obtained a median EPFG of 0.24 ± 0.012 , implying that encoding is ~24% stronger on preferred rather than anti-preferred phases.”

We thank the reviewer for cueing us to discuss the upward bias of the cosine fitting approach, which reminded us to further frame the strength of our method and to check the role of normalization in our findings.

Second, we make more explicit that the modeling studies of the Kopell group that showed that input spike phases can be retained in a local circuit for different input streams at different phases is not yet implying a functional feasibility/significance of such a code but ‘only’ a mechanistic possibility. First, we write:

“The parallel coding of information at a common beta rhythm in these models provides a qualitative proof of concept about phase specific encoding of multiple types of inputs in larger beta rhythmic ensembles, and suggests a possible mechanistic realization of enhanced encoding by the phase of firing in the beta band²³. “

Then we added an explicit discussion part to point to yet missing information by writing:

“Future work needs to specify whether these scenarios are realized by beta rhythmically firing ensembles of neurons and how long-lasting and robust the encoding with phase specific firing is with regard to the overall firing rates and firing variability of individual neurons during active brain states.”

Third, we directly adjusted the discussion (as suggested) to point out that the phase-based multiplexing that we describe in the discussion in only one possible scenario. We add to the discussion on the role of temporal coding the following section in the revised paper:

“... A similar caveat in interpreting the phase-modulated coding we report is that it is consistent with multiple coding schemes beyond a phase-based multiplexing (Akam and Kullman 2012). For example, spiking activity may be phase-synchronized in one condition but not another, or alternatively, conditions may be encoded on separate phases. Our results provide support for both coding schemes. Outcome cells resemble coding via an asynchronous code; that is to say, spike-LPF phase synchronization is higher in one condition than another, with no evidence of phase differences between conditions. On the other hand, RPE and outcome history cells show evidence of phase-separation coding. These cells showed no significant difference in PPC between conditions but did show a (near significant) trend towards firing on different phases.”

Fourth, we recognized that we had a couple of rather strong formulations about phase coding in the original submission that might seem as an overstatement of the functional importance of phase coding. To avoid this impression were adjusted the text at various places to more descriptively convey our results. For example, instead of writing e.g.

“... variables are encoded at separable phases of firing of neurons” we now write
“... variables are better encoded when taking into account the phase of firing of neurons that synchronize long-range across primate fronto-striatal circuits”

This change and various others are highlighted in **red font** in the revised text.

Comment 5. Basic characterisation of encoding:

The encoding regression analysis predicted neuronal firing as a function of recent outcomes (reward vs no reward) but does not take the choices made by the subject on these trials into account. This is a problem because the authors use the beta weights from this regression to infer whether neurons encode reward prediction errors (RPE), but clearly the influence of the previous trials reward on the current trials reward prediction error will depend on whether the previous trials choice is the same as the current trial choice. Indeed, the authors have previously done a much more in depth analysis of RPE coding in these brain regions on this task (Oemisch et al. Nat. Com. 2019), and showed that for the great major of RPE encoding neurons these RPEs were stimulus specific. More broadly, subjects cannot solve this task using outcome information alone – they need to know which stimuli the outcomes were associated with, so it is rather unsatisfying to see an analysis of learning related activity which ignores half the information needed to solve the task.

Reply: We would like to first clarify that our selection of learning variables takes explicitly into account the choices and choice history since our task rewarded correct choices deterministically. All *Outcome* encoding is thus indistinguishable from encoding correct vs erroneous choices and all reward history encoding is also choice history encoding. We agree very much that this outcome information needs to be combined with feature (color-) specific information, but we want to convey that this combination does not necessarily need to take place in individual neurons in order to help with learning (which is one reason we hope the current results would not be unsatisfying in that respect).

However, we fully agree that it would be parsimonious to have individual sites encoding the RPE selectively for only the reward relevant (color) feature, closely following the results and reasoning our previous work (Oemisch et al 2019). To address the reviewers suggestion, we therefore performed a preliminary analysis asking if there is feature specific RPE phase-encoding. We find that feature-specific phase-encoded RPEs are more prevalent than expected by chance. Specifically, we write:

“We next asked whether the EPFG for RPE encoding distinguishes the rewarded color that animals learned within a reversal block. Previously we showed that the firing rate of subsets of neurons encoded not only a scalar RPE signal but additionally showed stronger RPE signaling for only one or the other color in the task (Oemisch et al., 2019). These color-specific RPE signals can boost the reversal learning because they carry information not only about how much updating should take place (which scalar RPE’s signal) but the specific content of what needs to be updated (one or the other color during reversal learning). We quantified this feature-specific RPE encoding by separately testing whether the EPFG is significant when considering only trials when one or the other color was rewarded. We found that of all cell-LFP pairs encoding RPE’s, 3% (3/102) showed individually significant EPFG in both conditions (in other words, a non-feature-specific RPE), and ~15% (15/102) showed a significant EPFG only for one of two colors (a feature-specific RPE). The frequency of cell-LFP pairs where the EPFG was significant for neither, both, or only one color condition differed from chance (χ^2 test, $\chi^2=109$, $p=0$). Importantly, feature-specific EPFG was more common than feature non-specific EPFG ($\chi^2=6.72$, $p=0.01$). The proportion of color specific EPFG tended to be most prevalent in ACC with ~27% (9/34) of cases, compared to 10% (5/50) in LPFC and 6% (1/18) for STR (χ^2 test, $\chi^2=5.26$, $p=0.07$).”

We are grateful to the reviewer for having pointed us to this aspect. We are planning experiments with multiple features currently to do a more in-depth analysis on how much information can be extracted about relevant versus non-relevant features from the phase of firing (versus other codes). We have the impression that an in-depth analysis of feature specificity of coding (with clarifying the latencies, the area specificity, the frequencies specificity, and more) would make this already complex manuscript even more complex. In fact, we believe that this manuscript could serve as a good justification for future studies to quantify the amount of information and the limitations of coding through phase-specific firing.

Comment 6. Another question regarding the characterization of neurons is their clustering into three groups according to whether they represent current outcome, RPE or outcome history. This clustering plays a prominent role in subsequent analyses, so it is important that readers can clearly assess how distinct these putative clusters are. This is not clear because the clustering approach is rather unusual, making the metrics presented of cluster separation hard to interpret. An initial clustering was performed using a sensible distance metric (cosine distance on the regression beta weights). This clustering was repeated many times using bootstrap resampling of neurons, and a new similarity metric was created by assessing the number of times across the bootstrap samples each pair of neurons was clustered together.

This new similarity metric was used for a final clustering. As far as I can tell it is this new similarity metric that is presented in figure S1B to show the similarity between and across clusters, but this metric is so far removed from the neurons actual encoding of task variables that it is not informative about cluster separation. The authors should show the similarity matrix (as figure S1B) using the underlying cosine distance metric, and also show the location of individual neurons colored coded by cluster assignment in the space defined by the beta weights for current and previous trial outcome (or a similar space defined by the actual beta weights) so the cluster separation can be visualized.

Reply: We understand the reviewers concern that the similarity measure we use is many steps removed from the actual data. The reason we do so is because when clustering simply on the basis of cosine distance, we find that some neurons do indeed shift their cluster assignment. This is due to the sensitivity of k-means clustering to the initial cluster center assignment (Celebi and Kingravi 2012). Thus, we opted to use a method that clusters on the basis of cluster stability, similar to earlier works (Ardid et al 2015).

To actively address the reviewers' suggestion and help assessing the validity of our cluster assignment, we added the following analysis, results and text to the revised paper:

First, we added references in the methods section that directly support our clustering approach based on stability. We write:

“We clustered cells on the basis of their clustering stability⁸¹. We opted for this method because k-means clustering can be sensitive to initial conditions⁸². This involved three steps: (1) choosing the optimal number of clusters N_c , (2) measuring clustering stability, and (3) performing the final clustering.”

Second, we followed the reviewer suggestion and added more explicit data on the clustering and individual neuron assignments in eight new panels in a revised supplementary Figure S2. Figure S2A is the similarity matrix based on clustering stability (moved from Figure S1). Figure S2B is the similarity matrix based on the underlying cosine distance metric. Figure S2C quantifies the “goodness” of cluster assignment via the Silhouette measure for each encoding type and split by brain areas. Figure S2D plots the individual neurons colored coded according to their cluster assignment in a 2D space defined by their beta weights for the current and previous outcome. Figures SE-H compare different properties (such as encoding strength, firing rates, and effect timing) of these clusters. We refer to Figure S2 by writing:

*“We used a clustering analysis to test whether the three types of outcome encoding were separable from each other and prevalent in each of the recorded brain areas (**Figure 2E-G, Figure S2A,B**). Clustering showed that neurons encoding each the three variables were statistically separable with reliable cluster assignments of neurons evident in an average Silhouette measure of cluster separability of 0.81 for LPFC, 0.57 for ACC, and 0.75 for STR (**Figure S2C**)⁴⁰. The clustering does not preclude the possibility of a more continuous encoding space, but it statistically justifies focusing analysis on three sets of neurons with well distinguishable encoding pattern (see **Figure S2D**).”*

Comment 7. Other points:- When analyzing the behavior using a regression analysis (Figure 1D) it would be more informative to show how previous choices and outcomes jointly predict current choice (as in e.g. Parker ... Witten, Nat. Neurosci. 2016), rather than how previous outcomes predict current outcomes.

Reply: While we agree that this would be interesting, the task we employ is not appropriate for this sort of analysis. This is due to its deterministic nature. Thus, previous choices and rewards would provide redundant information in predicting the current choice.

Comment 8. - It was not completely clear to me how the encoding metrics defined in line 622 were used in evaluating EPG. I think that one metric was used for each neuron, with the metric chosen based on the neurons cluster assignment, but please clarify this.

Reply: Yes, this is right. We added this clarification in the methods:

“One encoding value was selected for each spike-LFP pair, on the basis of the cluster assignment of the spiking neuron.”

Comment 9. - To give readers a more rounded picture of how beta oscillation activity interacts with the behavioral task, it would be useful to show how beta power varies across trial epoch and with correct/incorrect outcomes, and whether there is any phase resetting of the oscillation by trial events.

Reply: We followed the suggestions and included in the revised manuscript more specific information:

We include time frequency plots and an analysis of the phase consistency (indicative of phase resetting) around the reward onset and the stimulus onset in a new supplementary Fig S8. We refer to it in the main text:

“... we found that band limited power in the beta band was a prominent and sustained component of the LFP after reward onset (Figure S8A) but without a reward-onset locked phase consistency (Figure S8B).”

Comment 10. - Line 34 seems to suggest that striatum ‘feeds back projections’ to cortex – presumably this was referring to indirect connections, but the wording is unclear.

Reply: We clarified that this is primarily via the thalamus and now write:

“Both regions project to partly overlapping regions in the anterior striatum (STR), which feeds back projections via the thalamus and thereby close recurrent fronto-striatal-thalamic loops.”

Comment 11. - Line 177 – for each neuron the frequency was selected that showed the greatest spike-LFP synchrony. How variable are these frequencies across neurons?

Reply: Thank you for pointing us to this part which we now quantify directly (finding that the consistency supports our findings and conclusions) and report the results in a new Suppl. Fig S11. We write that we:

*“... found that the three sets of neuronal encoding clusters synchronized on average at the same ~15 Hz center frequency (Kruskal Wallis, $\chi^2=0.95$, $p=0.62$; **Figure S11A**), and that they showed maximal phase-of-firing encoding at similar frequencies (also ~15Hz) (Kruskal Wallis test, $\chi^2=0.39$, $p=0.82$; **Figure S11B**).”*

Comment 12. - The authors term the phase dependent encoding they identify as phase-of-firing coding, but it would be more accurately termed a phase modulated rate code, because neurons still encode

behavioral variables by changes in their overall firing rate, but these changes are concentrated at particular phases.

Reply: We agree with the reviewer that what we report is more accurately described as a phase-of-firing modulation (or gain) of a rate code. To better reflect this, we have re-labeled our main encoding metric the Encoding Phase-of-Firing Gain (*EPFG*) and adjusted the terminology and wording across the entire revised manuscript (all changes made in red font).

Comment 13. - Figure 2-A: the raster plots do not add any value.

Reply: We understand this point but are also influenced by a large field of primate neurophysiologists for whom the raster's have remained an important visual sanity check. On this account, we believe it important to retain them.

Comment 14. – Line 70: ‘This phase-of-firing gain of encoding exceeded the firing rate code’ This is misleading as the depth of modulation of rate coding across the cycle was much smaller than the average encoding strength. I think what the authors are trying to convey is that taking spike phase into account increased information available to the readout compared to just reading the firing rates. However, to really quantify this the authors would need to use a decoding approach – i.e. asking how much information can be extracted by e.g. a linear classifier which either just has access to spike rates or has access to spike rates binned by phase. Such a decoding analysis would be a really nice complement to the encoding analysis and not a lot of work given that most of the preprocessing needed is already implemented.

Reply: We follow the reviewer and agree that our main results indicates that taking the phase into account enhances a firing rate code (but phase information alone does not exceed a rate code). In the original text we referred to that encoding gain at some places as information, which could be misleading because we did not perform a decoding analysis of the encoded information. To address the request for adding a decoding approach, we took recourse to an analysis of “explained deviance” (similar to ANOVA Percent Explained Variance) that previous nonhuman primate papers used and thus allows a comparison to those studies.

The method we applied (and added to the revised manuscript) (1) quantifies the percent deviance explained (which is more appropriate for the Poisson GLM model we consider here) for each phase bin, (2) fits a cosine to estimate the change in percent explained as a function of phase, and (3) compares it to a null distribution. We describe this approach in the revised methods section by writing:

“The EPFG effectively quantifies the difference in mean firing rates between conditions, as a function of LFP phase. However, this does not necessarily imply that the information is easily decodable by other brain circuits. To address this question, we asked how much variance can be explained by the model fit to data in each phase bin. To this end, for each fit on each phase bin, we extracted the percent deviance explained (analogous to the ANOVA percent variance explained but modified for a Poisson GLM). The percent deviance explained D^2 was calculated as⁸⁵:

$$D^2 = 1 - \left(\frac{\text{Residual Deviance}}{\text{Null Deviance}} \right)$$

The deviance for a Poisson distribution is defined as:

$$\text{Deviance} = 2 * \sum_i^n Y_i * \log\left(\frac{Y_i}{\lambda_i}\right) - (Y_i - \lambda_i)$$

Where Y_i is the observed spike count on trial i , and λ_i is the predicted spike count. We then determined how D^2 varied as a function of phase using the same procedure as described above; namely, we fitted a cosine to the D^2 of each phase bin, extracted the amplitude and phase, and compared it to a null-distribution where phases have been permuted. We call this quantity the Encoding Phase-of-Firing Gain (D^2), or EPFG $_{D^2}$.”

With this approach we obtained results that we visualize in a new supplementary Figure S7A and summarize in the revised results section by writing:

“The EPFG is an effect size measure for how strong firing rate is modulated by LFP phase between conditions. However, it does not take into account the variability of firing rates across trials, leaving open the question of whether such mean firing rate changes may be effectively decoded. To address this question, we performed additional tests at the same beta frequencies at which neurons maximally synchronized. Firstly, we calculated how much the percent explained deviance varied as a function of phase, which quantifies how well the model fit the data with spikes extracted on individual phase bins. We term this quantity EPFG $_{D^2}$ (see Methods). We found that across areas and all spike-LFP pairs with significant encoding, EPFG $_{D^2}$ was significantly larger than chance (Wilcoxon signrank test, $p \sim 0$). EPFG $_{D^2}$ was significantly above chance for Outcome (Wilcoxon signrank test, $p \sim 0$) and RPE ($p = 0.001$) clusters, but not for spike-LFP pairs with neurons from the Outcome History cluster ($p = 0.24$) (Figure S7A).”

Similarly, the results describing the phase at which encoding is maximal also remained the same, with RPE cells encoding on anti-preferred phases, whereas Outcome and Outcome History cells spanned the entire range of possible phases. These are visualized in a new Figure S7C, and described in the main text as:

“... RPE encoding neurons significantly encoded at similar phase-offsets relative to the neuron’s synchronizing phases (average phase: -2.76 ± 0.047 SE radians, $p = 0.0004$, corresponding to 27 ms away from the mean spike phase at a 15 Hz oscillation cycle), which was significantly different than the mean spike phase (Median test, $p = 0.027$). This effect was particularly pronounced for RPE cells in ACC (Figure S10), and was consistent across both monkeys (Figure S4C). Qualitatively similar results were obtained when extracting the preferred encoding phases derived from model deviances (Figure S7C).”

In summary, we extend our heartfelt thanks to the reviewer for having pointed us to several weaknesses that we hope to have addressed in a good manner that improved the validity, clarity and appeal of the manuscript.

REVIEWERS' COMMENTS:

Reviewer #1 (Remarks to the Author):

The revised manuscript addressed my concerns.

Reviewer #2 (Remarks to the Author):

I appreciate the very thorough revision, which I believe has further strengthened this interesting work. The authors adequately addressed all my previous concerns. I have no further comments.

Reviewer #3 (Remarks to the Author):

The authors have done a very thorough job of addressing my comments. I have no further substantive concerns, but have made a few minor comments/suggestions below. The study is very nice and I have no hesitation recommending it for publication.

- The authors describe the new permutation test which preserves within trial correlations by saying 'When constructing null distributions by randomly perturbing the phase of spikes on each trial'. It is not clear from this whether they perturb the phase of all spikes on a given trial by the same amount (which would preserve the within trial correlation structure) or each spike by a different amount (which would destroy it). From the context of their response I think it is the former, but this should be clarified in the text.

- Regarding the point in my first review that the depth of modulation found in the study may potentially underestimate the true modulation depth. I was actually trying to make a slightly different point from the question the authors addresses of whether the method used is biased to underestimate modulation depth in a statistical sense. Rather I was thinking about the fact that LFP is a very indirect measure of a population oscillation, because it combines signals from neurons participating in the oscillation with signals from other neuronal populations and any other noise sources. These will add noise to any estimate of oscillation phase obtained from the LFP, which will cause any estimate of modulation depth to be an underestimate. It is very hard to assess how large these effects might be but it seems plausible to me that they are substantial.

- I take the reviewers point that in their study outcome coding is identical to correct/incorrect coding and is therefore a relevant learning variable considered on it's own.

- The improved visualization of the clustering are very useful. However, panel S2D showing the clusters in a 2D space defined by current and previous outcome encoding is somewhat confusing because while in the clustering analysis the authors have (reasonably) flipped the sign of neurons that code current outcome negatively, they have not done this for the scatter plots, so each cluster is split into two when plotted.

- Data/code availability: The only statement I could find regarding data/code availability was 'Code will be made available upon request to the authors.' Good open science practice these days is to make both data and code needed to reproduce the manuscript figures available in a public repository. This is really fundamental to ensuring that work is reproducible, and the code provides a useful resource for other researchers wishing to build on the analyses. Please consider making the code and preferably the data available in a public repository (see <https://www.nature.com/nature-research/editorial-policies/reporting-standards> and <https://www.nature.com/news/announcement-where-are-the-data-1.20541>).

Point-by-point replies (in blue font) to comments of the Reviewer:

Reviewer #3 (Remarks to the Author):

Comment: The authors have done a very thorough job of addressing my comments. I have no further substantive concerns, but have made a few minor comments/suggestions below. The study is very nice and I have no hesitation recommending it for publication.

Reply: We are very grateful for the constructive and helpful comments of the reviewer which helped to considerably improve the manuscript. Thank you.

Comment:- The authors describe the new permutation test which preserves within trial correlations by saying ‘When constructing null distributions by randomly perturbing the phase of spikes on each trial’. It is not clear from this whether they perturb the phase of all spikes on a given trial by the same amount (which would preserve the within trial correlation structure) or each spike by a different amount (which would destroy it). From the context of their response I think it is the former, but this should be clarified in the text.

Reply: We have updated this section to read: “When constructing null distributions by randomly perturbing the phase of spikes on each trial by the same amount”.

Comment:- Regarding the point in my first review that the depth of modulation found in the study may potentially underestimate the true modulation depth. I was actually trying to make a slightly different point from the question the authors addresses of whether the method used is biased to underestimate modulation depth in a statistical sense. Rather I was thinking about the fact that LFP is a very indirect measure of a population oscillation, because it combines signals from neurons participating in the oscillation with signals from other neuronal populations and any other noise sources. These will add noise to any estimate of oscillation phase obtained from the LFP, which will cause any estimate of modulation depth to be an underestimate. It is very hard to assess how large these effects might be but it seems plausible to me that they are substantial.

Reply: We appreciate these considerations and have included two references to the revised text that convey that phase estimation critically depends on the level of background noise and synchrony in the neurons underlying the population response, writing:

“These suggestions depend on a proper estimation of phase, which can be influenced by the level of background noise⁷⁵ and the degree of synchrony of individual cells within a population⁷⁶. However, we believe this would not affect the main conclusions of our study, as we observed (1) significant increases in encoding phase gain both when oscillatory bursts were prominent or not, and (2) significant phase encoding gain when controlling for outcome induced activity.”

Comment: - I take the reviewers point that in their study outcome coding is identical to correct/incorrect coding and is therefore a relevant learning variable considered on it's own.

Reply: ok.

Comment: - The improved visualization of the clustering are very useful. However, panel S2D showing the clusters in a 2D space defined by current and previous outcome encoding is somewhat confusing because while in the clustering analysis the authors have (reasonably) flipped the sign of neurons that code current outcome negatively, they have not done this for the scatter plots, so each cluster is split into two when plotted.

Reply: We have updated the plot to reflect encoding preference after adjusting error encoding cells.

Comment: - Data/code availability: The only statement I could find regarding data/code availability was 'Code will be made available upon request to the authors.' Good open science practice these days is to make both data and code needed to reproduce the manuscript figures available in a public repository. This is really fundamental to ensuring that work is reproducible, and the code provides a useful resource for other researchers wishing to build on the analyses. Please consider making the code and preferably the data available in a public repository (see <https://www.nature.com/nature-research/editorial-policies/reporting-standards> and <https://www.nature.com/news/announcement-where-are-the-data-1.20541>).

Reply: We have included code that will fully allow to recreate all major analyses, and link to it online at: https://github.com/att-circ-ctrl/ana_phaseGain